# National quantifications of methane emissions from fuel exploitation using high resolution inversions of satellite observations

Lu Shen [1] ✉, Daniel J. Jacob [2], Ritesh Gautam[3], Mark Omara [3], Tia R. Scarpelli[4], Alba Lorente[2], Daniel Zavala-Araiza [3,5], Xiao Lu [6], Zichong Chen[2] & Jintai Lin [1]

Reducing methane emissions from fossil fuel exploitation (oil, gas, coal) is an important target for climate policy, but current national emission inventories submitted to the United Nations Framework Convention on Climate Change (UNFCCC) are highly uncertain. Here we use 22 months (May 2018-Feb 2020) of satellite observations from the TROPOMI instrument to better quantify national emissions worldwide by inverse analysis at up to 50 km resolution. We find global emissions of $62.7 \pm 11.5$ ($2\sigma$) Tg a$^{-1}$ for oil-gas and $32.7 \pm 5.2$ Tg a$^{-1}$ for coal. Oil-gas emissions are 30% higher than the global total from UNFCCC reports, mainly due to under-reporting by the four largest emitters including the US, Russia, Venezuela, and Turkmenistan. Eight countries have methane emission intensities from the oil-gas sector exceeding 5% of their gas production (20% for Venezuela, Iraq, and Angola), and lowering these intensities to the global average level of 2.4% would reduce global oil-gas emissions by 11 Tg a$^{-1}$ or 18%.

Methane ($CH_4$) is the second most important anthropogenic greenhouse gas after $CO_2$ and is responsible for 0.6°C global warming since preindustrial times[1]. Under the Paris Agreement, individual countries must set goals for mitigating their anthropogenic methane emissions relative to current baselines. The Global Methane Pledge signed by over 110 countries commits to reducing collective methane emissions by 30% by 2030[2]. Emission from fossil fuel exploitation (oil, gas, and coal) is an important mitigation target because it is estimated to account for about one-third of the global anthropogenic total and could be cost-effective to control[3–6]. National emission inventories submitted by individual countries to the United Nations Framework Convention on Climate Change (UNFCCC) under the Paris Agreement follow bottom-up approaches in which emission factors are applied to

activity data, sometimes with additional facility-specific information. But these national inventories are typically uncertain by a factor of two or more[6] and often by more than an order of magnitude for the oil-gas sector[5]. This uncertainty hinders the setting and tracking of mitigation goals for methane emissions.

Top-down approaches apply inverse methods to infer emissions from measurements of atmospheric methane. They use prior information from the bottom-up inventories and provide an independent way of improving these inventories. Many previous top-down studies have exploited methane observations from surface sites, aircraft, and the Greenhouse gases Observing SATellite (GOSAT) satellite[7–16] but these observations are very sparse. Satellite observations from the Tropospheric Monitoring Instrument (TROPOMI) launched in October

[1]Department of Atmospheric and Oceanic Sciences, School of Physics, Peking University, Beijing, China. [2]School of Engineering and Applied Sciences, Harvard University, Cambridge, MA 02138, USA. [3]Environmental Defense Fund, Washington DC 20009, USA. [4]School of GeoSciences, University of Edinburgh, Edinburgh EH9 3JN, UK. [5]Institute for Marine and Atmospheric Research Utrecht, Utrecht University, 3584 CC Utrecht, The Netherlands. [6]School of Atmospheric Sciences, Sun Yat-sen University, Zhuhai, Guangdong, China. ✉e-mail: lshen@pku.edu.cn

2017 provide considerably higher global data density, with continuous daily mapping of methane columns at 7 km × 5.5 km nadir resolution[17]. The TROPOMI observations have been applied to detect large point sources[18–20] and quantify emissions from a few source regions[21–24]. A global inversion showed artifacts in early versions of the data[25] that have since largely been corrected[17].

Here, we conduct a global ensemble of regional inversions of TROPOMI data to quantify emissions from fossil fuel exploitation worldwide at up to 50-km resolution, validate the results with field measurements across the globe, and further report improved inventory estimates for all countries in support of the Paris Agreement. Furthermore, we use our inversion framework to assess the sensitivity of the results to different bottom-up inventories, satellite data density, and choices of inversion parameters, from where we define the conditions under which TROPOMI can provide significant information to assist national bottom-up estimates.

## Results

### Correction factors to UNFCCC inventories and evaluation with field campaign data

We tile the world with 15 domains that account for 96% of global emissions from fossil fuel exploitation in 2019 according to the Global Fuel Emissions Inventory version 2 (GFEI v2), which allocates national

emissions reported to the UNFCCC on a 0.1°×0.1° grid[5]. Figure 1a shows these 15 domains and the gridded UNFCCC emissions. A few countries have not reported their emissions to the UNFCCC since 2000, notably Iraq and Libya, and for these GFEI v2 uses standard IPCC Tier 1 methods[5]. We conduct inversions of the TROPOMI observations in each domain from May 2018 to February 2020, using the GEOS-Chem model (https://doi.org/10.5281/zenodo.3634864) for simulation of atmospheric transport, to quantify emissions with a resolution of up to 0.5°×0.625° (~50km) for grid cells with significant fuel emissions (>1 Gg a⁻¹). We apply weight to the gridded bottom-up prior emission estimates for fuel and other sectors with the TROPOMI observations using analytical Bayesian optimization to solve for the maximum-likelihood posterior estimates of methane emissions on the 0.5°×0.625° grid (Eq. 1), and from there we aggregate emissions to the regional and country scales. Each grid cell includes emissions from nonfuel sectors as given by the respective prior inventories. Here, we attribute the posterior corrections to the different sectors in the grid cell on the basis of their error-weighted contributions to the prior emissions[23,24], and sectors with higher prior uncertainty are subject to larger relative corrections (Supplementary Note 1). This error weighting is particularly important for grid cells containing wetland emissions, which are particularly uncertain. We derive the global emissions from fossil fuel exploitation by summing posterior estimates for all 15 domains and

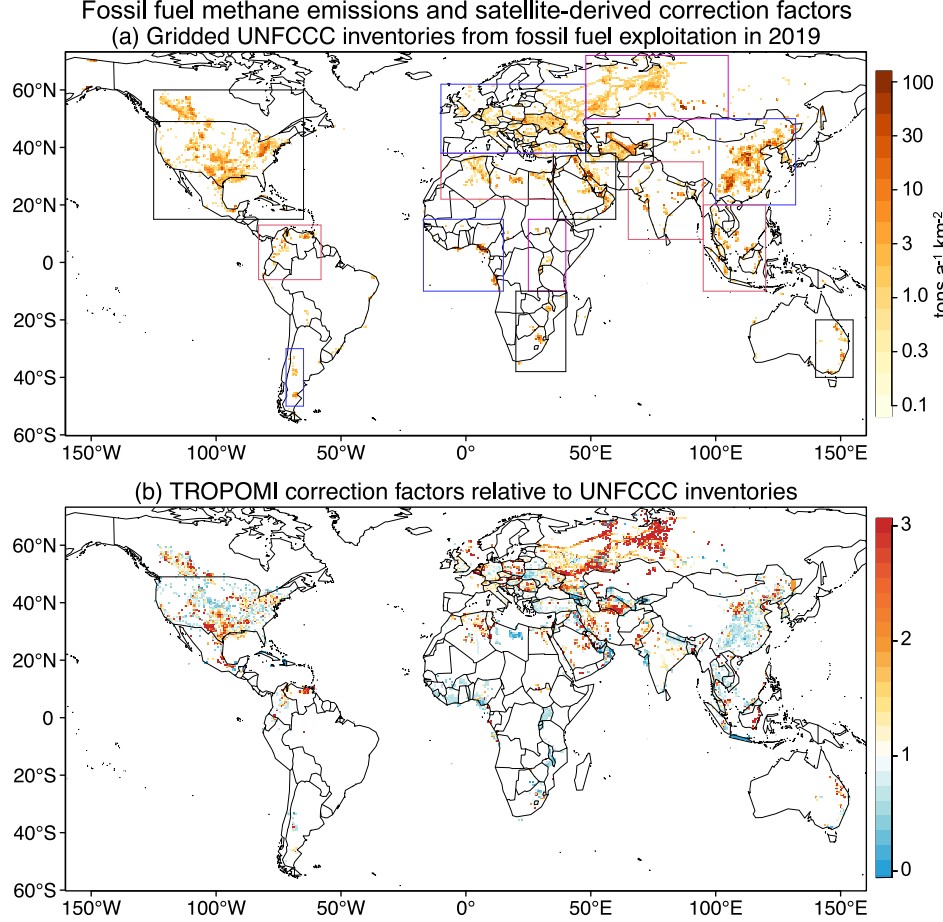

**Fossil fuel methane emissions and satellite-derived correction factors**
**(a) Gridded UNFCCC inventories from fossil fuel exploitation in 2019**

**(b) TROPOMI correction factors relative to UNFCCC inventories**

**Fig. 1 | Gridded UNFCCC (United Nations Framework Convention on Climate Change) inventories from fossil fuel exploitation and satellite-derived correction factors. a** Gridded GFEI v2 inventory of national emissions from fossil fuel exploitation (oil, gas, and coal) reported by individual countries to the UNFCCC for 2019 or the most recent year and used as the baseline prior estimate in our inversion of TROPOMI satellite observations. A few countries have not reported their emissions to the UNFCCC since 2000 (notably Iraq and Libya) and for these

GFEI v2 uses standard IPCC Tier 1 methods. We conduct the inversions for the 15 rectangular domains shown in the Figure, accounting for over 96% of global GFEI v2 emissions. **b** Satellite derived correction factors relative to the gridded UNFCCC inventories. Values are shown at the 0.5°×0.625° resolution of the inversion. The basemap is from the mapdata package (version 2.3.1) in R (https://cran.r-project.org/web/packages/mapdata/index.html).

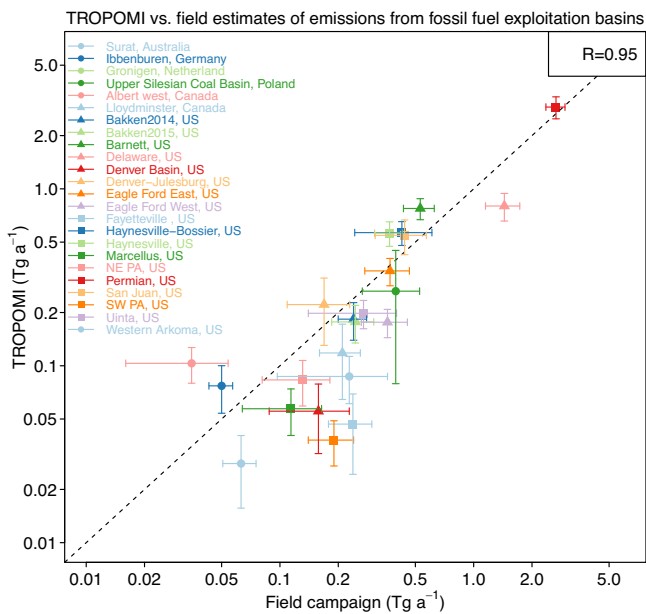

**Fig. 2 | Methane emissions from 24 oil-gas and coal production basins across the globe.** Estimates from field campaigns are compared to results from our TROPOMI inversions. For the US and Canada basins, we adjust TROPOMI's results to campaign years using relative linear trends of observation-derived (surface measurements + GOSAT) basin-scale emissions from Lu et al.[54]. The 1:1 line is dashed, and the correlation coefficient is shown inset. More details, including references for the field campaigns can be found in Supplementary Table 1-2. The error bars denote 95% confidence levels. Note the log-log scale.

adding the GFEI v2 inventory for the 2.8 Tg a$^{-1}$ of emissions outside of the domains (4% of global GFEI v2 emissions). We conduct an ensemble of inversions with different prior estimates (GFEI v2, GFEI v1, Emissions Database for Global Atmospheric Research version 6 or EDGARv6) and different assumptions on error statistics to quantify the uncertainty in our results. More details are provided in the Methods section.

Figure 1b shows the spatial distribution of optimized correction factors from fossil fuel exploitation relative to the gridded UNFCCC inventories. Corrections are upward in most large fuel production basins including in North America (the US, Canada, and Mexico), Venezuela, Russia, Central Asia (Turkmenistan and Kazakhstan), Africa (Algeria, Egypt), Middle East (Saudi Arabia, Iran), and Australia. Notable countries with downward corrections include Nigeria, Libya, and China.

Figure 2 compares our top-down emission estimates in large fossil fuel exploitation basins to 24 independent estimates of emissions from field campaigns, including in situ and remote sensing observations across the globe (Supplementary Table 1-2 for more details). The field campaigns were carried out over a range of time periods between 2011 and 2020. The correlation coefficient between our TROPOMI-derived posterior estimates and the field campaign estimates is 0.95 with no systematic bias. Results further indicate that TROPOMI can successfully detect emissions as low as 0.2 Tg a$^{-1}$ on the basin scale.

### Global emission estimates
Our optimized estimates of global methane emissions from TROPOMI data are 62.7 ± 11.5 Tg a$^{-1}$ for the oil-gas sector and 32.7 ± 5.2 Tg a$^{-1}$ for the coal sector, yielding a total fossil fuel emission of 95.4 ± 12.9 Tg a$^{-1}$ for the period of May 2018 to February 2020. Here and elsewhere, uncertainties are reported as two error standard deviations (2σ) from Monte Carlo analysis of our inversion ensemble using posterior error covariance matrices. These global estimates have low sensitivity to the priors used (Supplementary Fig. 1), implying that they are mainly determined by satellite observations. We evaluate our posterior

emission estimates by implementing them in GEOS-Chem and using them to simulate column-averaged methane mixing ratios for comparison with TROPOMI and GOSAT, and surface concentrations for comparison with surface measurements from the National Oceanic and Atmospheric Administration (NOAA) network[26]. Results show consistent improvements in the model-observation bias relative to using prior emissions (Supplementary Fig. 2-5).

Figure 3 compares our global emission estimates to previous literature in the context of their reporting periods going back to the 1980s and separating oil-gas from coal when available. Bottom-up estimates tend to show increases over the period, reflecting increasing production. The GFEI v2 inventory (based on 2019 UNFCCC reports) is much lower than other bottom-up estimates, largely driven by downward revision of Russian oil emissions reported to the UNFCCC[5]. Top-down fossil fuel estimates show large variability (80–110 Tg a$^{-1}$) but tend to be lower by 20 Tg a$^{-1}$ than bottom-up inventories (110–130 Tg a$^{-1}$, except GFEI v2), and comparison across top-down studies does not suggest a global increase of emissions over the 2010–2020 period (Fig. 3a).

Our satellite-based global estimate of oil-gas emissions (62.7 ± 11.5 Tg a$^{-1}$) for 2018–2020 is in the range of previous top-down studies (59–70 Tg a$^{-1}$) and 30% higher than GFEI v2 (48 Tg a$^{-1}$) (Fig. 3b), while our estimate of coal emissions (32.7 ± 5.2 Tg a$^{-1}$) is higher than previous top-down studies (20-30 Tg a$^{-1}$) and in agreement with GFEI v2 (33 Tg a$^{-1}$) (Fig. 3c) (details in Supplementary Table 3). The low coal emissions in previous top-down studies could stem from incorrect spatial distribution of Chinese emissions in older bottom-up inventories[27,28]. When combining oil-gas and coal emissions together, our top-down estimate (95.4 ± 12.9 Tg a$^{-1}$) is at the lower 20$^{th}$ percentile of the distribution of all previous estimates for 2010–2021(80–140 Tg a$^{-1}$). The next section attributes these differences to individual countries.

### National emission estimates and comparison to UNFCCC reports
Figure 4 shows our posterior estimates of oil-gas and coal methane emissions from the top emitting countries and compares to the UNFCCC reports. National data for all individual 93 countries with total fuel emissions larger than 1 Gg a$^{-1}$ are compiled in Supplementary Table 4. Posterior estimates of the top-20 oil-gas and coal emitting countries using different prior inventories can be found in Supplementary Table 5-6. Also shown in Fig. 4a is the methane intensity, defined as the national oil-gas emissions per unit of total gas production[4] (assuming 90% of methane content as in Alvarez et al.[4]), and the coal emission factors per unit production as defined by IPCC[29]. Emission factors for the individual oil and gas countries are given in Supplementary Fig. 6. Overall, our higher global oil-gas emission estimate relative to the UNFCCC is largely driven by underestimates of emissions from the US, Russia, Venezuela, and Turkmenistan in the national reports. Of these top emitters, Venezuela, Turkmenistan, Uzbekistan, Angola, Iraq, Ukraine, Nigeria and Mexico have methane intensities of 5-25% from the oil-gas sector (>20% for Venezuela, Iraq, and Angola). Lowering the emission intensity of these 8 countries to the global average level of 2.4% would reduce oil-gas methane emissions by 11 Tg a$^{-1}$ or 18% globally, implying a high margin for emission mitigation. China's oil-gas emission (2.7 ± 1.1 Tg a$^{-1}$) is twice higher than the UNFCCC inventory, consistent with Chen et al.[28], and these corrections are largest in the north (Fig. 1 and Supplementary Fig. 7) where a number of ultra-emitting point sources from the oil industry have been detected from TROPOMI data[20].

Our posterior estimate for Russia's oil-gas emissions is 9.4 Tg a$^{-1}$ but with large uncertainty (95% confidence level, 4.5-16.6 Tg a$^{-1}$) that reflects both the low observation density of TROPOMI at high latitudes (Supplementary Fig. 8-9) and a wide range in the prior inventories (Supplementary Fig. 10). Russia previously applied the IPCC 2006 emission factor for developing countries in its UNFCCC reports and this was used in GFEI v1, but its latest UNFCCC 2019 report used in GFEI

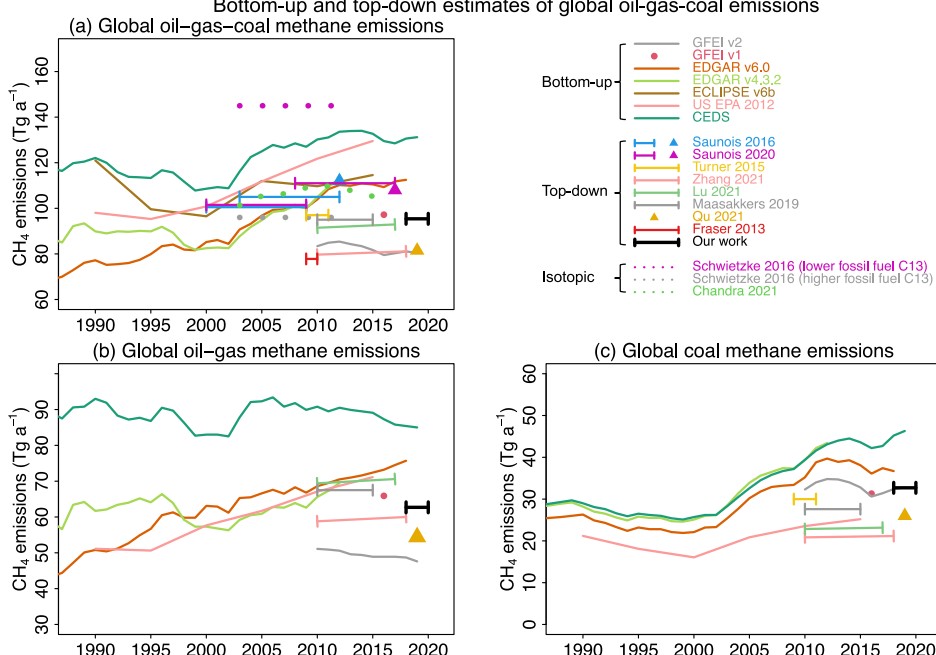

**Fig. 3 | Bottom-up and top-down estimates of global methane emissions from fossil fuel exploitation.** Results from our work (thick black segment) are compared to bottom-up inventories and to other top-down estimates for different years. Bottom-up inventories include GFEI v2[5] and GFEI v1[41], which are based on UNFCCC national estimates, EDGARv6.0[42], CEDS v2021[55], ECLIPSE version 6b[56], EDGAR v4.3.2[43], and US EPA 2012[57]. Top-down estimates include Zhang et al.[16], Lu et al.[15], Qu et al.[25], Maasakkers et al.[12], Saunois et al.[3,6], Turner et al.[9], and Fraser et al.[58]. Isotopic studies include Schwietzke et al.[59] and Chandra et al.[60]. (**a**) is for the total emissions from all fuel exploitation sectors. Fewer studies separate oil-gas and coal, shown in panels (**b**) and (**c**). Note differences in scales between panels. Top-down error estimates are either not reported or unrealistically low (<5%) (except Saunois et al.[3,6] and Fraser et al.[58]), which can be found in Supplementary Table 3 for more details.

v2 applies the emission factor for developed countries, resulting in a downward revision of its national oil-gas emissions by a factor of 5 (from 20.5 to 2.1 Tg a$^{-1}$ for oil and from 4.5 to 2.0 Tg a$^{-1}$ for gas)[5]. Overall, our results suggest that the most recent UNFCCC 2019 report is more accurate but needs to be corrected upward by a factor of two. Lauvaux et al.[20] found a large number of ultra-emitting point sources from the oil-gas sector in Russia including in particular from pipelines. Future satellite instruments may be more effective at observing high latitudes and Russian point sources[30].

Venezuela (posterior estimate of 4.0± 3.0 Tg a$^{-1}$) and Nigeria (1.5± 1.2 Tg a$^{-1}$) are the two largest oil-gas emitters in the tropics from our TROPOMI analysis, again with high uncertainty because of low TROPOMI observation density related to extensive cloudiness. Spatial co-location of oil-gas basins with wetlands in Nigeria further increases the difficulty of separating oil-gas emissions (Supplementary Fig. 11). Nigeria's emissions reported to UNFCCC increased from 0.4 (GFEI v1 for 2016) to 3.3 Tg a$^{-1}$ (GFEIv2 for 2019) after adopting an emission factor at the upper limit of the IPCC (2006) recommendations[31]. Our inversion implies that the more recent report should be reduced by 40-50%. The need for upward corrections of Venezuela's emissions has been previously reported in inversions of GOSAT data[12,15]. Lu et al.[15] estimated oil-gas emissions for Venezuela of 7.7 Tg a$^{-1}$ for 2010-2017. Our lower value (4.0 Tg a$^{-1}$) may be related to declining oil production over the 2016-2019 period as a result of intensified economic sanctions[5] (Supplementary Fig. 12).

Our posterior estimate of Turkmenistan's oil-gas emissions (3.6±1.3 Tg a$^{-1}$) is 2.4 times higher than its UNFCCC report (1.5 Tg a$^{-1}$), and makes it the 4$^{th}$ largest oil-gas emitter in the world (Fig. 4). This could be due to a large population of high-emitting point sources not accounted for in the bottom-up estimates. We find that the upward corrections are largest in the southern production basins (Supplementary Fig. 13), consistent with the previous high emitters identifications[32–34].

Figure 4b also displays the top 20 coal-emitting countries in our posterior estimates. Unlike for oil-gas emissions, we find in general good agreement with the UNFCCC inventories. This is likely because the country-scale emission factors from coal production have much lower variability than for oil-gas (Fig. 4) and emissions originate from a relatively small number of facilities. China's coal-based methane emission is 18.9 ± 3.3 Tg a$^{-1}$, slightly but not significantly lower than the UNFCCC inventory estimate. Our estimate for China is comparable to the range of 16.2-18.0 Tg a$^{-1}$ from recent satellite-based estimates (Supplementary Table 7)[28]. Indonesia is the second largest coal producer in the world but its submitted emission to the UNFCCC is only 0.2 Tg a$^{-1}$[15]; our posterior estimate is 0.7 Tg a$^{-1}$ but with very large uncertainty (0.1-2.7 Tg a$^{-1}$) because of extensive cloudiness and retrieval difficulties associated with coastlines (Supplementary Fig. 8). The largest relative correction factors are for Australia and Kazakhstan, where we find emissions to be respectively 1.8 and 4.8 times higher than in the UNFCCC reports. Our result for Australia is consistent with a recent finding that emissions from three large coal mines are seven times larger than the bottom-up estimates[19].

We find that the ability of TROPOMI to quantify fuel emissions for a given country varies greatly depending on the country, as illustrated in Fig. 4 and Supplementary Fig. 14. This variability is determined by the TROPOMI data density, the magnitude of national emissions, and the prior uncertainty in these emissions which we estimate on the basis of the range of the bottom-up inventories used in our inversion ensemble (GFEI v2, GFEI v1, EDGARv6). For the top fuel-emitting countries (>1Tg a$^{-1}$), the posterior/prior relative uncertainty reduction for total fuel emissions is 40% on average with a range from 10% to 70%, suggesting the effective role of TROPOMI in improving country-scale emissions. After combining inversions from the full ensemble (see Methods for more details), our results show that TROPOMI can constrain the emissions with a relative posterior uncertainty <30% (2σ) for most large mid-latitude emitters, including China, the US,

National emission estimates from TROPOMI inversion and UNFCCC reports

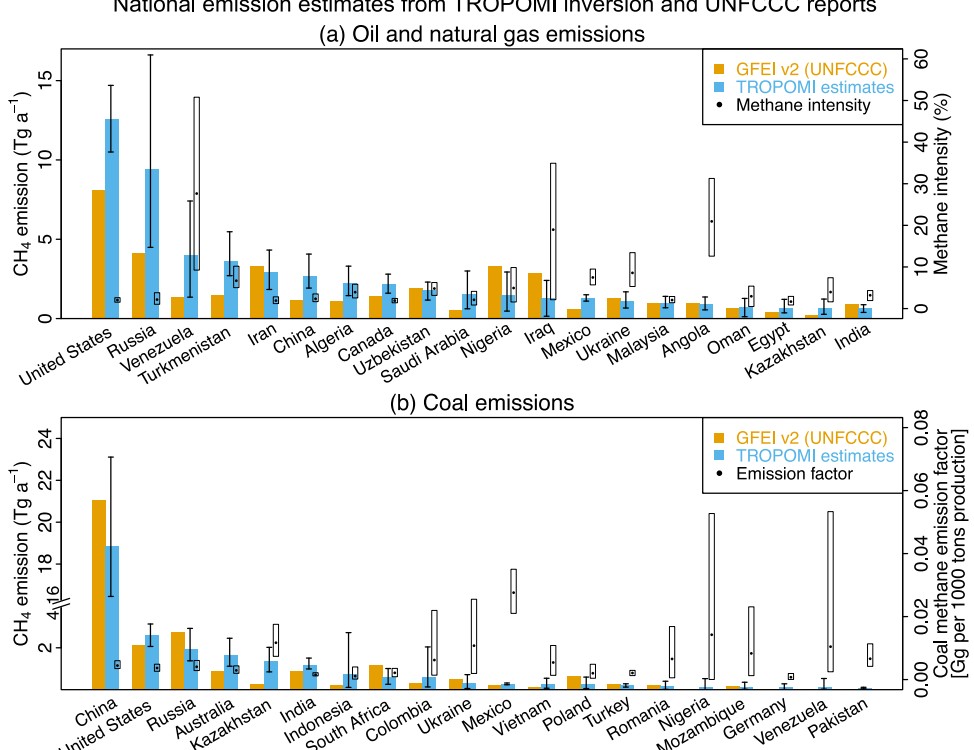

**Fig. 4 | National methane emissions from the (a) oil-gas and (b) coal sectors estimated by inversion of TROPOMI observations and compared to the UNFCCC reports.** The TROPOMI observations are for May 2018 – February 2020, and the UNFCCC reports are for 2019 (Annex I countries) or most recent (other countries), as compiled by the GFEI v2 inventory of Scarpelli et al.[5]. Iraq has not reported to the UNFCCC since 2000 and its emission is estimated in GFEI v2 using IPCC emission factors. The top 20 emitting countries are shown here; data for the 93 countries with total fuel emissions larger than 1 Gg a⁻¹ are in Supplementary

Table 4. Vertical bars indicate the 95% confidence levels from the inversion ensemble. The circles represent the methane intensity from oil-gas production (**a**), defined as the total oil-gas emission per unit of gas produced (assuming 90% methane content for gas)[4], and the coal emission factor (**b**), defined following IPCC[29] as the total coal emission per unit coal produced. The empty rectangles denote the 95% confidence levels of oil-gas emission intensities and coal-based emission factors. Note break in left ordinate axis of bottom panel, as Chinese coal emissions are much higher than for any other country.

Turkmenistan, Kazakhstan, India, and Canada. It has more difficulty in Russia and the tropics (Supplementary Fig. 14), where satellite observation density is relatively low and oil/gas fields are often collocated with wetlands, so that inversion results have limited information content and are sensitive to the choice of prior inventory (Supplementary Table 5-6).

## Discussion

An additional limitation in quantifying oil-gas emissions at high latitudes and in the tropics is the uncertainty in wetland emissions. Whereas oil-gas fields at northern mid-latitudes are usually in arid regions, oil-gas fields at high latitudes and in the tropics are often co-located with wetlands and separating the two emission sectors can be difficult. Our prior estimate of wetland emissions is obtained by averaging the 9 highest-performance members of the wetland methane emissions and uncertainty dataset for atmospheric chemical transport models (WetCHARTs v1.3.1)[35,36], for a global emission of 149 Tg a⁻¹ in 2019. We conducted another set of inversions using the ensemble average of all 18 members of WetCHARTs v1.3.1 with prior wetland emissions that are 10% higher globally, 20-30% higher in the tropics, and 15% lower in central Russia (Supplementary Fig. 15). The resultant changes in posterior fossil fuel emissions are 0.7 Tg in Venezuela and <0.2 Tg a⁻¹ in other countries (Supplementary Fig. 15).

There are other poorly accounted sources of uncertainty in our analysis. TROPOMI observations are affected by regional bias in some parts of the world[17,25] and the biases not removed by our quality flags (see Methods) would propagate to our inversion results. High latitudes and tropics have large seasonal variations in observation density that

would affect inversion results if fossil fuel emissions were seasonally variable (we assume that they are not). Independent inversions for different seasons show near-zero posterior corrections in the wintertime at high latitudes because of the low observation density (Supplementary Fig. 8-9, 16, Supplementary Note 2). GEOS-Chem transport error is treated as random through the observational error covariance matrix, but any systematic transport bias[37] would again propagate to inversion results. Our evaluation of inversion results with field campaigns in Fig. 2 does not reveal obvious biases but these campaigns are limited to North America and Europe. Offshore oil-gas emissions are not directly observed by TROPOMI in the source grid cell (TROPOMI observations over the oceans are limited to the glint mode, which we do not use here)[17] and are optimized solely on the basis of their plumes transported across coastlines and over land. Global offshore oil-gas emissions amount to 3.8 Gg a⁻¹ in GFEIv2 (7.9% of total oil-gas emissions) and we find that only 60% of these offshore emissions are effectively optimized in our inversion (averaging kernel sensitivity larger than 0.1).

In summary, we have used 22 months of TROPOMI satellite observations (May 2018 – February 2020) in an inverse analysis to quantify methane emissions from the fossil fuel industry (oil, gas, and coal) globally at up to 50 km resolution. We find that global methane emissions are 62.7 ± 11.5 Tg a⁻¹ from the oil-gas sector and 32.7 ± 5.2 Tg a⁻¹ from the coal sector, as compared to corresponding global estimates of 47.6 and 32.8 Tg a⁻¹ from the latest compilation of national bottom-up inventories reported to the UNFCCC. Our higher oil-gas estimate (by 30% or 15 Tg a⁻¹) is largely driven by underestimate of emissions from the US, Russia, Venezuela, and Turkmenistan in the

national reports. Eight countries have methane emissions from the oil-gas sector in excess of 5% of their gas production (>20% for Venezuela, Iraq, and Angola), and lowering their methane intensity to the global average level of 2.4% would reduce methane emission from the oil/gas sector by 11 Tg a$^{-1}$ or 18% globally, implying large emission mitigation potential. TROPOMI can quantify emissions with a relative posterior uncertainty <30% for most large mid-latitude emitters including China, the US, Turkmenistan, Kazakhstan, India, and Canada, but has more difficulty in Russia and the tropics where observations are less dense.

## Methods

### Satellite observations

We use the TROPOMI methane product version 2.02 from the Netherlands Institute for Space Research[17] for the period May 2018 - February 2020. TROPOMI is aboard the Sentinel 5 Precursor (S5-P) satellite and has a -13:30 local overpass time. It provides daily global coverage of methane columns at a spatial resolution of 7 km × 7 km (7 km × 5.5 km after August 2019)[38,39]. The methane column dry mixing ratio (XCH4) is retrieved from the sunlight backscattered by the Earth's surface in the shortwave infrared (SWIR) at 2305-2385 nm with near-unit sensitivity down to the surface under clear-sky conditions. The retrieval success rate is limited by cloud cover and by heterogeneous or dark surfaces, and is 3% globally over land[17]. We only use recommended high-quality XCH$_4$ measurements with the following criteria: (1) qa_value ≥ 0.5, (2) blended albedo ≤0.85, and (3) surface altitudes ≤ 2 km. The blended albedo, combining the surface albedo in the NIR and SWIR, can be used to filter scenes covered by snow[40]. The total number of TROPOMI observations is 1.4×10$^8$ for May 2018-February 2020, with large variability in observation density across the globe[30].

### Gridded national bottom-up inventories

We use the Global Fuel Exploitation Inventory version 2 (GFEI v2[5]) in 2019 as the baseline prior inventory for fossil fuel methane emissions. GFEI v2 is based on the national inventories reported to the UNFCCC for oil, gas, and coal, spatially allocated to 0.1°×0.1° resolution. We divide the globe into 15 inversion domains that can account for 96% of fossil fuel methane emissions based on this inventory. We also consider the GFEI version 1 inventory[41] for the year 2016, based on earlier UNFCCC reports, and the EDGARv6 inventory[42] for the year 2018 to evaluate the sensitivity of our results to the prior estimates (Supplementary Fig. 17). Such sensitivity arises from differences in both the magnitude and the spatial distribution of emissions among inventories, especially in regions like Russia, the Middle East, Venezuela, and Nigeria (Supplementary Fig. 18). Prior anthropogenic emissions from other sources including livestock, waste, and rice cultivation are from EDGAR v4.3.2[43]. Wetland emissions are taken from the mean of the nine high-performance members of the WetCHARTs v1.3.1 inventory ensemble[36]. Other natural sources include open-fire emissions from the Global Fire Emissions Database version 4s (GFED4s)[44], termite emissions from Fung et al.[45], and geological seepage emissions from[46] with global scaling to 2 Tg a$^{-1}$[47]. Details of sectorial contributions can be found in Supplementary Table 8.

### Forward model

We use the GEOS-Chem 12.7.0 chemical transport model (https://doi.org/10.5281/zenodo.3634864) as the forward model to simulate the sensitivity of atmospheric methane to emissions. GEOS-Chem is driven by MERRA2 reanalysis meteorological fields with 0.5°×0.625° horizontal resolution within the regional domains of Fig. 1, nested within a global simulation at 4°×5° resolution. Following Shen et al.[23], we ensure that model boundary conditions are consistent with local TROPOMI observations by scaling the GEOS-Chem vertical fields in each boundary grid square both temporally and spatially to match smoothed TROPOMI column observations. Methane sinks including atmospheric oxidation and soil absorption are included in our forward model but are not optimized because they are slow and relatively smooth.

### Construction of the state vector

The state vector for the inversion in individual regions consists of the spatially resolved emissions to be optimized on the 0.5°×0.625° model grid, along with boundary conditions for each quadrant. Our inversion mainly targets grid cells with high fossil fuel emissions, thus we can aggregate grid cells in areas of less interest, an approach often used by previous studies[23,24,48]. Following Shen et al.[24], our state vector includes native 0.5°×0.625° grid cells where prior fuel emissions exceed 1 Gg a$^{-1}$ and aggregates grid cells elsewhere. Altogether, the state vector consists of 5651 elements globally. The inversion optimizes total methane emissions for the state vector grid cells including contributions from all sectors. We follow Shen et al.[23] to attribute the posterior correction factors and variances in each grid cell to individual sectors based on the error-weighted contributions from these sectors in the prior inventory.

### Atmospheric inversion

We perform the inversion analysis following our previous work in North America[23,24]. The optimized methane emissions are obtained by minimizing a Bayesian cost function that balances the information from observations and the prior emissions, weighed by the corresponding error covariances[49]. For this we assemble gridded emissions and boundary conditions into a state vector $\mathbf{x}$, and the May 2018-February 2020 TROPOMI methane column data into an observation vector $\mathbf{y}$. Applying Bayes' theorem and assuming Gaussian errors leads to an optimized maximum-likelihood estimate for $\mathbf{x}$ by minimizing the cost function J given by

$$J(\mathbf{x}) = (\mathbf{x} - \mathbf{x_A})^T \mathbf{S_A}^{-1}(\mathbf{x} - \mathbf{x_A}) + \gamma(\mathbf{y} - \mathbf{Kx})^T \mathbf{S_O}^{-1}(\mathbf{y} - \mathbf{Kx}) \qquad (1)$$

Here $\mathbf{x_A}$ is the vector of prior emissions and boundary conditions, $\mathbf{K} = \partial\mathbf{y}/\partial\mathbf{x}$ is the Jacobian matrix describing the sensitivity of methane columns to the perturbation of every element in $\mathbf{x}$, and $\mathbf{S_A}$ and $\mathbf{S_O}$ are covariance matrices for prior and observational errors, both taken to be diagonal. For $\mathbf{S_A}$, we assume 50% error standard deviations for emission elements in the baseline inversion and 5 ppb error standard deviation for the boundary conditions. For $\mathbf{S_O}$, we apply the residual error method[50,51] by calculating the residual standard deviation between observations and GEOS-Chem simulations that use the prior estimates $\mathbf{x_A}$. The regularization term $\gamma$ is designed to account for unresolved observational error covariances in the inversion and thus avoid overfit to observations[52]. Following Lu et al.[15], we choose $\gamma$ such that $(\hat{\mathbf{x}} - \mathbf{x_A})^T \mathbf{S_A}^{-1}(\hat{\mathbf{x}} - \mathbf{x_A}) \approx n$ where $n$ is the number of state vector elements and is the expected value of the chi-square distribution for a diagonal matrix. This yields $\gamma$ in the range 0.01-0.2 for different inversion domains. Uncertainties in the specifications of $\mathbf{S_A}$ and $\gamma$, and in the assumption of Gaussian errors, are addressed with the inversion ensemble described below.

Minimization of Eq. 1 at $\nabla_x J(\mathbf{x}) = 0$ yields the optimal estimate $\hat{\mathbf{x}}$ for the state vector, the corresponding posterior error covariance matrix $\hat{\mathbf{S}}$, and the averaging kernel matrix $\mathbf{A}$ as follows

$$\hat{\mathbf{x}} = \mathbf{x_A} + \left(\gamma\mathbf{K}^T\mathbf{S_O}^{-1}\mathbf{K} + \mathbf{S_A}^{-1}\right)^{-1}\gamma\mathbf{K}^T\mathbf{S_O}^{-1}(\mathbf{y} - \mathbf{Kx_A}) \qquad (2)$$

$$\hat{\mathbf{S}}^{-1} = \gamma\mathbf{K}^T\mathbf{S_O}^{-1}\mathbf{K} + \mathbf{S_A}^{-1} \qquad (3)$$

$$\mathbf{A} = \mathbf{I} - \hat{\mathbf{S}}\mathbf{S_A}^{-1} \qquad (4)$$

where $\mathbf{I}$ is the identity matrix. The averaging kernel matrix $\mathbf{A}$ defines the sensitivity of the solution to the true state. The trace of $\mathbf{A}$ quantifies

the degrees of freedom for signal (DOFS), representing the number of independent pieces of information that can be effectively constrained in the inversion[53]. We construct the Jacobian matrix explicitly column-by-column by conducting 22-month sensitivity GEOS-Chem simulations perturbing individual state vector elements. This enables analytical solution for $\hat{\mathbf{x}}$ in Eq. 2 and closed-form expressions for $\hat{\mathbf{S}}$ and $\mathbf{A}$ following Eq. 3 and 4. Our inversion can constrain 568 pieces of independent information in the global spatial distribution of methane emissions (Supplementary Fig. 19).

Running a series of 15 regional inversions to cover the important source domains for methane (Fig. 1) is much less computationally expensive than a global inversion, and this allows for much higher spatial resolution. In addition, use of regional GEOS-Chem model simulations with TROPOMI boundary conditions regularizes the inversion by avoiding bias in initializing the model, accounting for the methane sink, and simulating stratospheric transport[48]. It also mitigates the effect of any large-scale regional biases in the TROPOMI data. The 15 inversion domains cover 96% of global oil-gas emissions, 82% of all anthropogenic emissions, and 51% of natural emissions according to our prior emission inventories (Supplementary Table 8). To calculate the world's total emissions, we sum posterior emissions for all 15 domains and retain prior emissions for the rest of the world. Our posterior estimate for global anthropogenic emission is 363 Tg a$^{-1}$, consistent with a global inversion of GOSAT and TROPOMI observations by Qu et al.[25].

### Inversion ensemble and uncertainty analysis

The posterior error covariance matrix $\hat{\mathbf{S}}$ represents the uncertainty within our choices of inversion parameters, but there is uncertainty in these parameters. Here we take advantage of the construction of the Jacobian matrix $\mathbf{K}$ to obtain a large ensemble of analytical solutions ($\hat{\mathbf{x}}$, $\hat{\mathbf{S}}$) by varying the parameters and prior estimates in the inversion. The ensemble has 39 members, including (1) use of either GFEI v2, GFEI v1, or EDGARv6 as prior bottom-up inventories, (2) regularization factors $\gamma$ varied by a factor of 0.5 and 2 from the baseline values; (3) error standard deviation of 50% and 75% on the prior estimate; (4) lognormal error statistics (geometric standard deviation of 2) for the prior estimate. We then use the Monte Carlo method to estimate the posterior error statistics from the ensemble according to ($\hat{\mathbf{x}}$, $\hat{\mathbf{S}}$) for each ensemble member. We report error statistics on the inversion results as two standard deviations ($2\sigma$), corresponding to the 95% confidence level.

## Data availability

The TROPOMI methane product is from https://doi.org/10.5281/zenodo.4447228. All data generated in this study are available at https://doi.org/10.18170/DVN/PRSYW1.

## Code availability

The code for the GEOS-Chem 12.7.0 chemical transport model is available at https://doi.org/10.5281/zenodo.3634864.

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

## Acknowledgements

Lu Shen was supported by the National Natural Science Foundation of China (4227050413). Work at Harvard was supported by the International Methane Emissions Observatory (IMEO) of the United Nations Environmental Programme (UNEP), by the Global Methane Hub, and by the NASA Carbon Monitoring System (80NSSC18K0178). Work at EDF was supported by the Robertson Foundation.

## Author contributions

L.S., D.J.J. and R.G. designed the experiments and L.S. carried them out. D.J.J. and R.G. helped supervise the project. A.L. provided satellite observations and T. S. provided the GFEIv2 inventory. M.O and D.Z.-A. contributed to the data analysis. X.L. and Z.C. contributed to the Bayesian inversion method. J.L. contributed to the policy implication. L.S. and D.J.J. prepared the manuscript with contributions from all co-authors.

## Competing interests

The authors declare no competing interests.
