## [Peer Review File · Nature Communications]

Worldwide quantification of national methane emissions from fuel exploitation using high resolution inversions of satellite observationsREVIEWER COMMENTS

Reviewer #1 (Remarks to the Author):

This study uses 22 months of TROPOMI satellite observations to quantify national fossil fuel emissions worldwide by inverse analysis techniques at finer (~ 50 km) resolution. This is of interest to the scientific community to understand the CH₄ budget and the policymakers in defining the baseline emissions to mitigate climate change. I have no expertise in some statistical methods (inverting XCH₄ from satellite) underpinning several calculations and therefore cannot evaluate them. Considering the pervasive amount of work behind the manuscript, the quality of the scientific approach and conclusive elements brought by the authors is significant. However, some extra effort is needed to improve the clarity of the current presentation and some missing details. Please find details of these points.

(1) How did you calculate the global emissions? Did you sum all the inverted emissions over 15 boxes? But as you mentioned that these boxes contain 96% of global emissions. How did you include 4% in estimating the global emissions?

(2) The details on the inversion are not very clear. Did you separately run the inversion at 15 boxes individually or simultaneously? Did you use a nested grid over each box? Why didn't you perform a single global inversion at each grid? How will the results be changed if you perform a single global inversion at each grid? How many sources did you optimize? A summary of other prior and optimized sources is good to report. What is the resolution of your forward model?

(3) What was the approach to select the best posterior estimate (Line 64-66)?

(4) How did you use the EDGARv6 inventory emissions for sensitivity for May 2018-Feb, 2020, when inventory emissions are available until 2018? Did you extrapolate the EDGARv6 inventory emissions? A clarification of these details is worth including.

(5) Some of the highest correction (e.g., over Russia) is collocated with the high wetland emissions. Further, the number of samplings in a different month (high solar zenith angle) differs over the higher latitude region (e.g., Russia). How robust are total emissions over the high latitude countries for uneven sampling in different seasons and collocated potent emission sources? A spatial plot of the total number of samplings in each season (for 2019 and 2020) may be good to show in the supplement.

(6) Most of the campaigns shown in Figure 2 are mainly in USA and Canada. More validation work is needed, especially the spatial allocation of the corrected correction in emissions. Checking the optimized model simulations with the meridional gradient (for 2019 and 2020) based on surface CH₄ observation (also the satellite observation) will be a nice check.

Technical Details:

Title: Good to replace "fuel exploitation" with fossil fuel exploitation"

Line 58-59: "... which allocates....as of September 2021." The message is not very clear. Consider rewriting.

Line 68. Use EDGARv6 consistently here and in other places.

Reviewer #2 (Remarks to the Author):

GENERAL COMMENTS

The manuscript by Shen et al. uses 2018-2020 data from the TROPOMI methane product and column observations to perform atmospheric inversion analyses using the GEOS-Chem 12.7.0 chemical transport model. Results are then compared to fossil fuel exploration basins, bottom-up, top-down, and isotopic emission inventories for the oil, gas, and coal industries. The study finds that emissions from the oil and gas industry are 30% higher compared to global total emissions from UNFCCC reports. This difference is mainly driven by underestimation by UNFCCC reports of the four largest emitting countries in the world: U.S., Russia, Venezuela, and Turkmenistan.

The manuscript is well written, organized, and with high-quality analyses. The results are based on widely used top-down and bottom-up data, established optimization methods, and robust statistical data analyses. The authors made a considerable effort to compare their results with a vast compendium of field campaigns and emission inventories. This manuscript will be a valuable contribution to scientific and policy purposes after revisions.

SPECIFIC COMMENTS

Line 67. Include a brief explanation of how error-weighted contributions were estimated.

Lines 87-88. Given that Fig S1 shows up to ~10% differences among posterior totals depending on the prior used, I suggest changing the phrase to: "global estimates have low sensitivity to the priors used".

Additionally, Fig. 4 shows large differences between UNFCCC reports depending on the country analyzed. Can a similar comparison of priors and posteriors be done for the top emitting countries? Such analysis would provide a disaggregated perspective of the sensitivity to the priors used.

National emission estimates and comparison to UNFCCC reports section. Fig.4 shows Mexico as the country with the highest coal methane emission factor, what is the uncertainty associated with the TROPOMI observations?

Materials and Methods general comments

Lines 140-141. The aggregation of grid cells based on prior emissions can be a good strategy for regional studies (like the ones cited here) where emission sources are well known. However, for a global scale study, such a strategy can mask important emission sources that are not spatially allocated correctly. This is especially true when using EDGAR as a prior inventory since multiple studies have shown important differences in spatial distribution between it and local inventories.

Lines 253-254. Why were lognormal error statistics performed only on GFEIv2 and not the other ones?

The inability of TROPOMI to detect offshore emissions is never mentioned in the methods section. What is the contribution to country-wide and global totals of unaccounted offshore oil and gas emissions?

MINOR EDITS

Line 68. Add the version of EDGAR used.

Lines 96-97. Add citations to previous top-down studies.

Reviewer #3 (Remarks to the Author):

Review of Shen et al., "Worldwide quantification of national methane emissions from fuel exploitation using high-resolution inversions of satellite data", for Nature Communications.

This work performs regional inversion analyses of 22 months of TROPOMI total-column methane measurements at a very high (50 km) spatial resolution to quantify coal and oil+gas fossil fuel emissions of methane to the atmosphere. This is critically important and valuable work, given the recent importance assigned to methane as a greenhouse gas, given that we have more power now to cut down our methane emissions than our CO₂ emissions in the short term. The fact that they report results (separately for oil+gas and coal) for roughly 100 countries/regions is nothing short of amazing, especially given some of the small areas of these countries. They also compare the oil+gas emissions to total production in order to quantify the "methane intensity" of each country, that is, the so-called "methane leakage rate". Therefore, after addressing the following comments and concerns, I suggest this work be published with all due speed.

"What are the noteworthy results?"

Their primary results are their unprecedented disaggregation of country-scale fossil fuel-based methane emissions over the time period of study. This is very noteworthy and of both high scientific and political/policy-based importance.

"Will the work be of significance to the field and related fields?"

Yes.

"How does it compare to the established literature?"

I am not a methane emissions researcher (currently), so it is difficult for me to fully answer this question. Based on the manuscript, it seems their results for various regions (Russia, China, etc) are generally consistent with previous findings.

"If the work is not original, please provide relevant references."

It certainly appears to be original.

"Does the work support the conclusions and claims, or is additional evidence needed?"

Usually, but not everywhere, as discussed in my comments below.

"Are there any flaws in the data analysis, interpretation and conclusions? Do these prohibit publication or require revision?"

Major flaw is the relatively incomplete error analysis as I discuss more fully below. This requires some level of revision.

"Is the methodology sound? Does the work meet the expected standards in your field?"

Yes.

"Is there enough detail provided in the methods for the work to be reproduced?"

It appears so, but I am not an inversion expert so I am not sure.

High-Level Comments:

My central critique of this work is the relatively incomplete nature of the error analysis discussion. Given the relatively high scientific and political importance of their findings, having appropriate and valid error bars on their results is critical. First, it appears based on the "Materials and Methods" discussion that the uncertainty due to wetland emissions is not included in their posterior emissions estimate. It seems to be it must be, as TROPOMI only sees the total CH₄ emissions, even in a relatively small area. For instance, what if the entire ensemble of WetCHARTS estimates had been

included, instead of just the mean? For many countries/areas this may not have been a significant contributor to the uncertainty, but certainly in some areas (especially in the tropics) it would be. The authors must discuss this potentially important source of uncertainty and why it is justified to completely ignore it.

Second, what is the contribution of atmospheric transport uncertainty? I realize they are doing regional inversions but still, given that they are trying to disaggregate coal, oil+gas, livestock, agricultural, and wetland emissions, I would think that atmospheric transport uncertainty could play a large role in their posterior emissions estimates. Is it feasible to perform these runs with a different transport model and compare the differences? If not, the authors MUST be clear from the outset the potential role of atmospheric transport uncertainty in potentially invalidating some of their conclusions.

Finally, I suggest including additional error bars in some of their figures, as discussed below.

Specific comments:

- I don't love the title. "Fuel exploitation" is not a common phrase. Suggest "fossil fuel exploitation" to be more specific.
- Figure 2: Please consider using a log-log scale (rather than linear) to better show the agreement at the lower emission rates.
- Figure 3: Please include vertical error bars for the various top-down measurements. These are a critical part of the measurement and are typically displayed on these plots, so we can get a sense of how well the different estimates agree to within their uncertainties, and how your uncertainty (rather than just absolute value of emissions) compares. If older Top-Down uncertainty estimates are artificially low because they do not include certain contributors to uncertainty, it is important for you to add this to your discussion.
- Figure 4: I like that you included emissions uncertainties on the TROPOMI estimates. But it is also very important to include them on the methane intensities (top plot) and emission factors (bottom plot). Please do this. (If not possible for some reason, please state why not in the text, and speculate on what your uncertainties on these values might be.)
- L117: Discussion of the large uncertainty on Russia's oil-gas estimate. Your data density for Russia is not much different for the U.S. it appears. It appears that the driving factor is the large disparity between GFEI v1 as compared to v2 and EDGARv6. Please comment on ways to potentially reduce this uncertainty, given the extremely important implications for these results?
- L128 area. I notice you do not comment on the ~factor of 2 discrepancy between GFEIv2 and TROPOMI for Nigeria's emissions. Please add a sentence on this, whether you think that any statements on this discrepancy are unwarranted given the relatively large uncertainty on your estimate, or if your value agrees better with GFEIv2 vs. EDGAR and possibly why.
- L131: regarding your comment on the "lower value (4.1 Tg a⁻¹) may be related to..." you should caveat this by saying ", though it is difficult to assess trends in their emissions given our the large uncertainty." Also, do the bottom-up inventories imply declining production in Venezuela during this time period, consistent with your statement? If so, say so!
- L133: "Is three times higher" → "Is 2.5 times higher" (3900/1500=2.6).
- L140: "China's methane emission is" → "China's coal-based methane emissions are".
- L141: ", lower by 10% than the UNFCCC inventory estimate" → ", consistent with the UNFCCC inventory estimate within our uncertainty." I don't think the claim that China's emissions are 10% lower is warranted given your large (17%) uncertainty.

Grammatical comments:

- L37: "which can provide" → "can provide"
- L63: "We weigh" → "We weight"
- L69: "More details are in" → "More details are provided in"
- L114: China's oil-gas emissions () are two times higher than the...

- L155: → "...can constrain the emissions with a relative posterior uncertainty of better than 30% (2σ)..."

Reviewer #4 (Remarks to the Author):

The authors use satellite observations of methane to estimate global methane emissions from fossil fuel extraction. Overall, I think the results of this manuscript are important, novel, and worthy of publication in Nature Communications. I also believe the manuscript is clear and well-written. I have included several suggestions for the manuscript below.

* The manuscript and SI do not include any model-data comparisons. I think that including model-data comparisons, particularly against independent CH₄ observations, is an important step for evaluating the inverse model. A handful of recent studies have found substantial differences between CH₄ emissions estimated using TROPOMI observations and GOSAT observations (see below), potentially due to errors in the TROPOMI CH₄ observations. These findings make model-data comparisons against independent observations all the more important. There are numerous independent datasets that could be used in this study for evaluation -- GOSAT observations, in situ observations from the NOAA Greenhouse Gas Reference Network, and observations from some of the campaigns cited in Fig. 2.

* A recent study by Liang et al. (<https://acp.copernicus.org/preprints/acp-2022-508/>) find differences of ~30% between inverse estimates of CH₄ for India and China that use TROPOMI observations versus GOSAT observations. Furthermore, they argue that emissions estimated using GOSAT observations are more consistent with surface observations. Liang et al. also use the updated Lorente et al. (2021) data product. By contrast, line 44 of the present manuscript argues that biases in the TROPOMI observations have largely been corrected. How do you reconcile the results of Liang et al. with the present study? Should those results imply larger uncertainty bounds on the reported CH₄ emissions estimated from TROPOMI?

* Materials and Methods section: How does the inverse model treat temporal variability in the emissions? The inverse model optimizes the overall magnitude and spatial patterns in the emissions, but does the inverse model optimize seasonal or year-to-year variability in the emissions? In other words, does the inverse model assume that the temporal distribution of emissions is known or fixed to the temporal distribution of the prior?

* Lines 18, 49, and 161: I find the wording in these lines somewhat misleading. The inverse model, as described in the Materials and Methods section, is used to estimate emissions at 50km resolution in selected grid boxes of the model domain. By contrast, the authors aggregate model grid boxes into coarser regions for other areas of the model domain. Hence, I think it's fair to say that the inverse model is used to estimate emissions at spatial resolutions up to 50km resolution, but the emissions are not optimized at that resolution everywhere.

Lines 36-38: This sentence is grammatically complex, and I think simplifying or breaking up the sentence would improve readability.

* Lines 46-47: I also find these lines to be somewhat misleading, and I recommend deleting this sentence from the manuscript. It is true that existing, global inverse modeling studies often estimate emissions at coarse spatial resolutions. With that said, the present study also makes compromises on resolution to improve the computational feasibility of the inverse model. For example, the emissions in the present study are estimated at coarser, aggregate resolutions in grid boxes where fossil fuel

emissions are less than 1 Gg a⁻¹. In addition, my impression is that the inverse model in the present study does not correct temporal variability in the emissions. Overall, the state vector in this study has 5651 elements, fewer elements than many global GHG inverse modeling studies that use satellite data.

Line 76: Independent estimates of what?

Response to referee comments on “Worldwide quantification of national methane emissions from fossil fuel exploitation using high-resolution inversions of satellite data”

We thank the four referees for their careful reading of the manuscript and valuable comments. This document is organized as follows: the Referee’s comments are in *italics*, our responses are in plain text, and all the revisions in the manuscript are shown in blue. **Blue text** here denotes text written in direct response to the Referee’s comments. The line numbers in this document refer to the updated manuscript **with tracked changes**.

Overall, we have made the following changes to improve our manuscript.

1. We have evaluated our posterior emission estimates by implementing them in GEOS-Chem and using them to simulate column-averaged methane mixing ratios for comparison with TROPOMI and GOSAT, and surface concentrations for comparison with in-situ observations in NOAA. Results show consistent improvements in the model-observation bias relative to using prior emissions. Please see Fig. S2-S5 and related texts for more details.
2. We have tested the effects of using different wetland priors on posterior estimates of fossil fuel methane emissions. Overall, we find the effects are larger (0.7 Tg a^{-1} or 17%) in Venezuela but tiny elsewhere. Please see Fig. S15 for more details.
4. We have added one paragraph (at the end of the Section ‘national emission estimates and comparison to UNFCCC reports’) to discuss the potential shortcomings of our method.
3. We have included many new figures and tables in supplementary materials to provide more details of our work. Please check.

Reviewer #1

This study uses 22 months of TROPOMI satellite observations to quantify national fossil fuel emissions worldwide by inverse analysis techniques at finer (~50 km) resolution. This is of interest to the scientific community to understand the CH₄ budget and the policymakers in defining the baseline emissions to mitigate climate change. I have no expertise in some statistical methods (inverting XCH₄ from satellite) underpinning several calculations and therefore cannot evaluate them. Considering the pervasive amount of work behind the manuscript, the quality of the scientific approach and conclusive elements brought by the authors is significant. However, some extra effort is needed to improve the clarity of the current presentation and some missing details. Please find details of these points.

Response. We thank the referee for making these valuable comments. Please see our following response.

(1) How did you calculate the global emissions? Did you sum all the inverted emissions over 15 boxes? But as you mentioned that these boxes contain 96% of global emissions. How did you include 4% in estimating the global emissions?

Response. Thanks for pointing this out. Now we make it clear in the main text how we calculate global emissions. Line 92. We derive the global emissions from fossil fuel exploitation by summing posterior estimates for all 15 domains and **adding the GFEI v2 inventory for the 2.8 Tg a⁻¹ of emissions outside of the domains (4% of global GFEI v2 emissions).**

Line 619. To calculate the world's total emissions, we sum posterior emissions for all 15 domains and **retain prior emissions for the rest of the world.**

(2) The details on the inversion are not very clear. Did you separately run the inversion at 15 boxes individually or simultaneously? Did you use a nested grid over each box? Why didn't you perform a single global inversion at each grid? How will the results be changed if you perform a single global inversion at each grid? How many sources did you optimize? A summary of other prior and optimized sources is good to report. What is the resolution of your forward model?

Response: The referee is right that we separately run the inversion at 15 boxes individually, which can save a lot of computational cost than running global inversions at a high resolution (~50km). For example, a global 2-year run at 50km resolution can take 15-20 days using 8 CPU cores, compared to <1 day in a regional domain like Europe. And we use nested simulations. Now we make it clear in the text.

Line 490. **Running a series of 15 regional inversions to cover the important source domains for methane** (Fig. 1) is much less computationally expensive than a global inversion, allowing for much higher spatial resolution.

Line 432. GEOS-Chem is driven by MERRA2 reanalysis meteorological fields with **0.5°×0.625°** horizontal resolution within the regional domains of Fig. 1, **nested within a global simulation at 4°×5° resolution.**

Theoretically, the results should be consistent if we perform a single global inversion at each grid because the Jacobian sensitivity won't change. Meanwhile, regional inversions have the advantage of avoiding the potential influence of regional bias in satellite data. Now we say this in text:

Line 491. In addition, use of regional GEOS-Chem model simulations with TROPOMI boundary conditions **regularizes the inversion by avoiding bias in initializing the model**, accounting for the methane sink, or simulating stratospheric transport. **It also mitigates the effect of any large-scale regional biases in the TROPOMI data.**

We optimize all sources inside our 15 inversion domains. Now we report all the emission changes of other prior and optimized sources (Table S8).

Line 494. **The 15 inversion domains cover 96% of global oil-gas emissions, 82% of all anthropogenic emissions, and 51% of natural emissions according to our prior emission inventories (Table S8). To**

calculate the world's total emissions, we sum posterior emissions for all 15 domains and retain prior emissions for the rest of the world. Our posterior estimate for global anthropogenic emission is 363 Tg a⁻¹, consistent with a global inversion of GOSAT and TROPOMI observations by Qu et al.²⁵.

Table S8. Posterior estimates of all sectors and comparison with another global TROPOMI inversion work.

	This study			Qu et al. ³⁰	
	Prior estimate	Percentage of prior emission covered by our 15 inversion domains	Posterior estimates	TROPOMI inversion	TROPOMI-GOSAT joint inversion
Total sources	516	71%	556	556	570
Anthropogenic	325	82%	363	336	363
Oil and Gas	48	96%	61	53	56
Coal	33	96%	33	NA	NA
Livestock	115	71%	127	126	139
Rice	37	86%	42	NA	NA
Wastewater	38	80%	42	44	44
Landfill	29	87%	31	27	31
Other anthropogenic sources	25	83%	27	23	26
Natural	191	51%	193	220	207
Wetlands	162	50%	165	195	183
Termites	12	56%	13	12	12
Open fires	15	61%	13	11	10
Seeps	2	74%	2	2	2

(3) What was the approach to select the best posterior estimate (Line 64-66)?

Response. Thanks for pointing out this. The best posterior estimate is the maximum likelihood estimate based on the Bayesian cost function. We realize the word ‘best’ here may be misleading; thus, we use “the maximum-likelihood posterior estimates” instead. And we refer to Eq. 1 in the main text to reduce confusion.

Line 85. We weight the gridded bottom-up prior emission estimates for fuel and other sectors with the TROPOMI observations using analytical Bayesian optimization to solve for **the maximum-likelihood posterior estimates** of methane emissions on the 0.5°×0.625° grid (**Equation 1**).

Line 456. Applying Bayes’ theorem and assuming Gaussian errors leads to an optimized **maximum-likelihood estimate** for x by minimizing the cost function J given by

$$J(x) = (x - x_A)^T S_A^{-1} (x - x_A) + \gamma (y - Kx)^T S_0^{-1} (y - Kx) \quad (1)$$

(4) How did you use the EDGARv6 inventory emissions for sensitivity for May 2018-Feb, 2020, when inventory emissions are available until 2018? Did you extrapolate the EDGARv6 inventory emissions? A clarification of these details is worth including.

Response. Thanks for pointing this out. Now we make it clear in the text.

Line 419. We also consider the GFEI version 1 inventory⁴⁰ **for the year 2016**, based on earlier UNFCCC reports, and the EDGAR v6 inventory⁴¹ **for the year 2018** to evaluate the sensitivity of our results to the prior estimates.

Line 416. We use the Global Fuel Exploitation Inventory version 2 (GFEI v2⁵) **in 2019** as the baseline prior inventory for fossil fuel methane emissions.

(5) Some of the highest correction (e.g., over Russia) is collocated with the high wetland emissions. Further, the number of samplings in a different month (high solar zenith angle) differs over the higher latitude region (e.g., Russia). How robust are total emissions over the high latitude countries for uneven sampling in different seasons and collocated potent emission sources? A spatial plot of the total number of samplings in each season (for 2019 and 2020) may be good to show in the supplement.

Response. Thanks for making this good point. The referee is correct that wetlands could potentially influence our results. As mentioned in our original manuscript, we used the 9 highest-performance members from WetCHARTs v1.3.1 as the prior inventory. As a sensitivity test, we used all 18 members from WetCHARTs to conduct another set of inversions. Now we discuss it in text as follows.

Line 312. An additional limitation in quantifying oil-gas emissions at high latitudes and in the tropics is the uncertainty in wetland emissions. Whereas oil-gas fields at northern mid-latitudes are usually in arid regions, oil-gas fields at high latitudes and in the tropics are often co-located with wetlands and separating the two emission sectors can be difficult. Our prior estimate of wetland emissions is obtained by averaging the 9 highest-performance members of the WetCHARTs v1.3.1 inventory ensemble^{38,39}, for a global emission of 149 Tg a⁻¹ in 2019. We conducted another set of inversions using the ensemble average of all 18 members of WetCHARTs v1.3.1 with prior wetland emissions that are 10% higher globally, 20-30% higher in the tropics, and 15% lower in central Russia (Fig. S15). The resultant changes in posterior fossil fuel emissions are 0.7 Tg in Venezuela and <0.2 Tg a⁻¹ in other countries (Fig. S15).

Effects of wetland emissions on posterior methane emissions from fossil fuel exploitation

(a) WetCHARTs wetland methane emissions (high performance – all ensembles)

(b) Δ methane emissions from fossil fuel exploitation in the top 5 countries (high performance – all ensembles)

	Venezuela	Russia	India	Nigeria	Cote d'Ivoire
Posterior OG and Coal emissions using high performance WetCHART members (Tg a ⁻¹)	4.2	11.3	1.9	3.4	0.27
Δ emissions if using all WetCHART members (Tg a ⁻¹)	-0.7	0.3	-0.2	-0.15	-0.15

Fig. S15. Effects of wetland inventories on posterior fossil fuel emissions. (a) Difference in wetland methane emissions using the 9 highest-performance members and 18 all members in WetCHARTs v1.3.1^{3,5}. The five inversion domains with the highest relative difference in wetland methane emissions are shown as black rectangles. (b) The 5 top countries with the highest absolute changes of posterior methane emissions from fossil fuel exploitation if using all 18 WetCHARTs members.

About the uneven sampling in different seasons, we now report the spatial plot of the total number of samplings and corresponding seasonal correction factors in the high latitudes. We have made the following revisions:

Line 321. There are other poorly accounted sources of uncertainty in our analysis. TROPOMI observations are affected by regional bias in some parts of the world^{17,25,37} and the biases not removed by our quality flags (see Methods) would propagate to our inversion results. **High latitudes and tropics have large seasonal variations in observation density that would affect inversion results if fossil fuel emissions**

were seasonally variable (we assume that they are not). Independent inversions for different seasons show near-zero posterior corrections in the wintertime at high latitudes because of the low observation density (Fig. S8-S9, S16, Text S2).

Text S2. Seasonal posterior correction factors and effects of data density at high latitudes

We also calculated posterior emissions from fossil fuel exploitation using TROPOMI observation in different seasons at northern high-latitudes, where the observation density varies considerably across one year (Fig. S8-S9). In these high-latitude countries, posterior corrections are nearly zero in winter because of scarce observations and low averaging-kernel sensitivities (Fig.S16). This result also demonstrates that seasonal corrections are partly influenced by satellite data availability, which does not necessarily reflect the temporal variability in emissions in regions with uneven observation density. In this study, we do not try to optimize for higher temporal variability of emissions because uneven and inadequate seasonal sampling frequencies are present in most regions of the world (Fig. S8-S9).

Figure S8. (a) TROPOMI data density (counts km⁻² a⁻¹) from May 2018 to February 2020, mapped to 0.5°×0.625° horizontal resolution. (b) Same as (a) but only for gridcells with fossil fuel methane emissions greater than 1 Gg a⁻¹.

Figure S9. Fraction of TROPOMI observation in different seasons (DJF, MAM, JJA and SON), mapped to $0.5^\circ \times 0.625^\circ$ horizontal resolution. Gridcells with zero observation in each season are shown as white.

Fig. S16. Posterior correction factors in different seasons at high latitudes north of 30°N . This test uses GFEIv2 as the prior inventory. Please note that the seasonal corrections here are largely determined by the sampling frequency in different seasons (Fig. S8-S9), which do not necessarily reflect the temporal variability in emissions. See Text S2 for discussion.

(6) Most of the campaigns shown in Figure 2 are mainly in USA and Canada. More validation work is needed, especially the spatial allocation of the corrected correction in emissions. Checking the optimized model simulations with the meridional gradient (for 2019 and 2020) based on surface CH_4 observation (also the satellite observation) will be a nice check.

Response. We thank the referee for making this good point. The referee is correct that most campaigns are in the US and Canada. To our knowledge, the campaign data in Fig 2 are all we can collect from recent publications. Now we acknowledge this shortcoming in the main text.

Line 328. Our evaluation of inversion results with field campaigns in Fig. 2 does not reveal obvious biases but these campaigns **are limited to North America and Europe**.

And we conduct more validation work in this revised manuscript.

Line 143. We evaluate our posterior emission estimates by implementing them in GEOS-Chem and using them to simulate column-averaged methane mixing ratios for comparison with TROPOMI and GOSAT, and surface concentrations for comparison with surface measurements from the National Oceanic and Atmospheric Administration (NOAA) network²⁶. Results show consistent improvements in the model-observation bias relative to using prior emissions (Fig. S2-S5).

Fig. S2. Bias of column-averaged methane mixing ratio in the prior and posterior GEOS-Chem simulations relative to TROPOMI from May 2018 to February 2022. To be consistent with our inversion setups, we run the GEOS-Chem model in 15 inversion domains using prior and posterior inventories at the $0.5^\circ \times 0.625^\circ$ resolution, and then aggregate the simulation results. The mean bias and root mean square errors (RMSE) are shown inset.

Fig. S3. Bias of column-averaged methane mixing ratio in the prior and posterior GEOS-Chem simulations relative to GOSAT from May 2018 to February 2022. To be consistent with our inversion setups, we run the GEOS-Chem model in 15 inversion domains using prior and posterior inventories at the $0.5^\circ \times 0.625^\circ$ resolution, and then aggregate the simulation results. The mean bias and root mean square errors (RMSE) are shown inset.

Fig. S4. Latitudinal column-averaged methane mixing ratio between GOSAT, TROPOMI, GEOS-Chem posterior runs. GEOS-Chem simulations are performed in 15 inversion domains with boundary conditions calibrated to match TROPOMI (same as our inversion setups). Thus, GEOS-Chem's latitudinal variability closely resembles TROPOMI's.

Fig. S5. Bias of surface methane concentrations in the prior and posterior GEOS-Chem simulations relative to NOAA in-situ observations inside our 15 inversion domains from May 2018 to February 2022 (ObsPack, Schuldt et al.⁴). The mean bias and root mean square errors (RMSE) are shown inset.

Technical Details:

Title: Good to replace “fuel exploitation” with fossil fuel exploitation”

Response. Thanks, now we use ‘fossil fuel exploitation’ in the main text.

Line 58-59: “.... which allocates....as of September 2021.” The message is not very clear. Consider rewriting.

Response. Thanks. The information of the year here is redundant. Now we say:

Line 78. We tile the world with 15 domains that account for 96% of global emissions from fossil fuel exploitation in 2019 according to the Global Fuel Emissions Inventory version 2 (GFEI v2), which allocates national emissions reported to the UNFCCC on a $0.1^\circ \times 0.1^\circ$ grid⁵.

Line 68. Use EDGARv6 consistently here and in other places.

Response. Fixed, thanks!

Reviewer #2

GENERAL COMMENTS

The manuscript by Shen et al. uses 2018-2020 data from the TROPOMI methane product and column observations to perform atmospheric inversion analyses using the GEOS-Chem 12.7.0 chemical transport model. Results are then compared to fossil fuel exploration basins, bottom-up, top-down, and isotopic emission inventories for the oil, gas, and coal industries. The study finds that emissions from the oil and gas industry are 30% higher compared to global total emissions from UNFCCC reports. This difference is mainly driven by underestimation by UNFCCC reports of the four largest emitting countries in the world: U.S., Russia, Venezuela, and Turkmenistan.

The manuscript is well written, organized, and with high-quality analyses. The results are based on widely used top-down and bottom-up data, established optimization methods, and robust statistical data analyses. The authors made a considerable effort to compare their results with a vast compendium of field campaigns and emission inventories. This manuscript will be a valuable contribution to scientific and policy purposes after revisions.

Response. We thank the referee for making these valuable comments. Please see our following responses.

SPECIFIC COMMENTS

Line 67. Include a brief explanation of how error-weighted contributions were estimated.

Response. Thanks, now we add this brief explanation to the text.

Line 90. and sectors with higher prior uncertainty are subject to larger relative corrections (Text S1).

Lines 87-88. Given that Fig S1 shows up to ~10% differences among posterior totals depending on the prior used, I suggest changing the phrase to: “global estimates have low sensitivity to the priors used”.

Response. That is corrected, thanks.

Additionally, Fig. 4 shows large differences between UNFCCC reports depending on the country analyzed. Can a similar comparison of priors and posteriors be done for the top emitting countries? Such analysis would provide a disaggregated perspective of the sensitivity to the priors used.

Response. We now have new Tables S5-S6 to provide these details. Please check. Now we say this in the main text.

Line 191. Posterior estimates of the top-20 oil-gas and coal emitting countries using different prior inventories can be found in Table S5-S6.

Line 275. After combining inversions from the full ensemble (see Methods for more details), our results show that TROPOMI can constrain the emissions with a relative posterior uncertainty <30% (2σ) for most large mid-latitude emitters including China, the US, Turkmenistan, Kazakhstan, India, and Canada. It has more difficulty in Russia and the tropics (Fig. S14), where satellite observation density is relatively low and oil/gas fields are often collocated with wetlands, so that inversion results have limited information content and are sensitive to the choice of prior inventory (Table S5-S6).

Table S5. Posterior estimates of oil-gas methane emissions (2018-2020) using different priors (GFEIv2, GFEIv1 and EDGARv6) for top emitting countries.

Countries	using GFEIv2 (UNFCCC)	using GFEIv1	using EDGARv6
-----------	-----------------------	--------------	---------------

	Prior (Tg a ⁻¹)	Posterior (Tg a ⁻¹)	Prior (Tg a ⁻¹)	Posterior (Tg a ⁻¹)	Prior (Tg a ⁻¹)	Posterior (Tg a ⁻¹)	Averaged posterior (Tg a ⁻¹)
United States [#]	8.1	12.6	8.3		9.6		12.6
Russia	4.1	5.6	24.9	15.0	6.9	7.7	9.4
Venezuela	1.4	4.1	3.2	6.4	0.9	1.6	4.0
Turkmenistan	1.5	3.9	1.5	3.8	1.3	3.1	3.6
Iran	3.3	2.2	4.1	2.6	7.4	3.9	2.9
China	1.2	2.2	1.1	2.1	3.2	3.8	2.7
Algeria	1.1	2.0	1.2	2.1	2.1	2.5	2.2
Canada [#]	1.4	2.2	1.6		1.6		2.2
Uzbekistan	1.9	2.1	2.6	2.1	1.2	1.1	1.8
Nigeria	3.3	2.1	0.4	0.5	3.7	1.9	1.5
Iraq	2.9	1.7	0.1	0.2	4.6	2.0	1.3
Saudi Arabia	0.6	0.8	0.6	1.1	4.7	2.8	1.5
Mexico [†]	0.6	1.3	0.6		0.9		1.3
Ukraine	1.3	1.3	1.1	0.9	0.9	1.1	1.1
Malaysia	1.0	0.9	0.9	0.9	1.3	1.2	1.0
Angola	1.0	0.8	1.1	1.0	1.2	0.9	0.9
Oman	0.7	0.9	0.1	0.3	1.1	0.9	0.7
India	0.9	0.7	1.0	0.8	0.7	0.5	0.6
Kazakhstan	0.2	0.5	0.3	0.6	0.9	0.9	0.7
Egypt	0.4	0.5	0.4	0.5	1.1	1.1	0.7

[#]Here the prior inventory is based on GFEIv2, which has improved spatial distribution relative to GFEIv1 and EDGARv6. We also scale the prior in the Permian basin to match EDF's inventory²⁰ (extrapolated from site-scale emission rates) in sensitivity experiments, following the same setups in Shen et al.³⁶. Thus, we do not use GFEIv1 and EDGARv6 as priors.

[†]Here we use GFEIv2 as the prior and scale Mexico's offshore emissions by a factor of 0.1 to match field campaign results³⁷, following the same setups in Shen et al.¹. So we do not use GFEIv1 and EDGARv6 as priors.

Table S6. Posterior estimates of coal-based methane emissions (2018-2020) using different priors (GFEIv2, GFEIv1 and EDGARv6) for top emitting countries.

Countries	using GFEIv2 (UNFCCC)		using GFEIv1		using EDGARv6		Averaged posterior (Tg a ⁻¹)
	Prior (Tg a ⁻¹)	Posterior (Tg a ⁻¹)	Prior (Tg a ⁻¹)	Posterior (Tg a ⁻¹)	Prior (Tg a ⁻¹)	Posterior (Tg a ⁻¹)	
China	21.1	20.2	18.5	18.2	19.9	18.2	18.9
United States [#]	2.1	2.6	2.9		1.6		2.6
Russia	2.7	2.0	2.5	1.8	3.1	2.0	2.0
Australia	0.9	1.6	1.0	1.7	0.9	1.6	1.6
Kazakhstan	0.3	1.2	0.9	1.4	0.7	1.4	1.3
India	0.9	1.1	0.8	1.0	1.2	1.4	1.2
Colombia	0.3	0.7	0.3	0.6	0.2	0.4	0.6
South Africa	1.2	0.7	0.4	0.5	1.2	0.7	0.6
Indonesia	0.20	0.19	0.14	0.13	4.51	1.93	0.75
Ukraine	0.48	0.30	0.65	0.36	0.16	0.24	0.30
Mexico [†]	0.23	0.22	0.22		0.04		0.22
Poland	0.66	0.30	0.79	0.29	0.44	0.16	0.25
Turkey	0.27	0.27	0.22	0.21	0.12	0.12	0.20
Viet Nam	0.11	0.13	0.09	0.09	0.50	0.60	0.27
Romania	0.22	0.24	0.24	0.20	0.03	0.03	0.16
Venezuela	0.04	0.17	0.02	0.04	0.03	0.11	0.10
Mozambique	0.18	0.18	0.02	0.03	0.18	0.17	0.13
Nigeria	0.00	0.00	0.25	0.43	0.00	0.00	0.14
Germany	0.01	0.01	0.10	0.14	0.14	0.18	0.11
Pakistan	0.04	0.06	0.06	0.09	0.04	0.05	0.07

[#]Here the prior inventory is based on GFEIv2, which has improved spatial distribution relative to GFEIv1 and EDGARv6. We also scale the prior in the Permian basin to match EDF's inventory²⁰ (extrapolated from site-scale emission rates) in sensitivity experiments, following the same setups in Shen et al.³⁶. Thus, we do not use GFEIv1 and EDGARv6 as priors.

[†]Here we use GFEIv2 as the prior and scale Mexico's offshore emissions by a factor of 0.1 to match field campaign results³⁷, following the same setups in Shen et al.¹. So we do not use GFEIv1 and EDGARv6 as priors.

National emission estimates and comparison to UNFCCC reports section. Fig.4 shows Mexico as the country with the highest coal methane emission factor, what is the uncertainty associated with the TROPOMI observations?

Response. We now have updated Fig. 4 to show the uncertainty associated with the emission intensity.

National emission estimates from TROPOMI inversion and UNFCCC reports

Fig. 4. National methane emissions from the oil-gas and coal sectors estimated by inversion of TROPOMI observations and compared to the UNFCCC reports. The TROPOMI observations are for May 2018 – February 2020, and the UNFCCC reports are for 2019 (Annex I countries) or most recent (other countries), as compiled by the GFEI v2 inventory of Scarpelli et al. ⁵. Iraq has not reported to the UNFCCC since 2000 and its emission is estimated in GFEI v2 using IPCC emission factors. The top 20 emitting countries are shown here; data for the 93 countries with total fuel emissions larger than 1 Gg a⁻¹ are in Table S2. Vertical bars indicate the 95% confidence levels from the inversion ensemble. The circles represent the methane intensity from oil-gas production (top panel), defined as the total oil-gas emission per unit of gas produced (assuming 90% methane content for gas) ^{4,29}, and the coal emission factor (bottom panel), defined following IPCC ³⁰ as the total coal emission per unit coal produced. **The empty rectangles denote the 95% confidence levels of oil-gas emission intensities and coal-based emission factors.** Note break in left ordinate axis of bottom panel, as Chinese coal emissions are much higher than for any other country.

Materials and Methods general comments

Lines 140-141. The aggregation of grid cells based on prior emissions can be a good strategy for regional studies (like the ones cited here) where emission sources are well known. However, for a global scale study, such a strategy can mask important emission sources that are not spatially allocated correctly. This is especially true when using EDGAR as a prior inventory since multiple studies have shown important differences in spatial distribution between it and local inventories.

Response. We thank the referee for this comment. Now we discuss it in the main text and have a new figure in the SI to support the discussion.

Line 421. Such sensitivity arises from **differences in both the magnitude and the spatial distribution of emissions among inventories**, especially in regions like Russia, the Middle East, Venezuela, and Nigeria (Fig. S18).

Line 277. It has more difficulty in Russia and the tropics (Fig. S14), where satellite observation density is relatively low and oil/gas fields are often collocated with wetlands, so that **inversion results have limited information content and are sensitive to the choice of prior inventory** (Table S5-S6).

Fig. S18. Spatial distribution of fossil fuel methane emissions from GFEIv2, GFEIv1 and EDGARv6. We only show those gridcells with emissions greater than 1 Gg a⁻¹.

Lines 253-254. Why were lognormal error statistics performed only on GFEIv2 and not the other ones?

Response. Now we apply that to all prior inventories, and we have updated all numbers and related figures.

Figure S1. Global prior vs. posterior methane emissions of fossil fuel exploitation using different prior inventories. Different colors represent the results using different bottom-up priors and assumptions of errors. Triangles denote the magnitude of the priors, **points refer to estimates assuming a lognormal error**, and the boxplots denote the distribution of the posteriors (see Methods for more details).

The inability of TROPOMI to detect offshore emissions is never mentioned in the methods section. What is the contribution to country-wide and global totals of unaccounted offshore oil and gas emissions?

Response. Thanks for this good point. Now we say this in text.

Line 329. Offshore oil-gas emissions are not directly observed by TROPOMI in the source grid cell (TROPOMI observations over the oceans are limited to the glint mode, which we do not use here)¹⁷ and are optimized solely on the basis of their plumes transported across coastlines and over land. Global offshore oil-gas emissions amount to 3.8 Gg a⁻¹ in GFEIv2 (7.9% of total oil-gas emissions) and we find that only 60% of these offshore emissions are effectively optimized in our inversion (averaging kernel sensitivity larger than 0.1).

MINOR EDITS

Line 68. Add the version of EDGAR used.

Response. Corrected, thanks.

Lines 96-97. Add citations to previous top-down studies.

Response. Updated, thanks.

Reviewer #3

Review of Shen et al., "Worldwide quantification of national methane emissions from fuel exploitation using high-resolution inversions of satellite data", for Nature Communications.

This work performs regional inversion analyses of 22 months of TROPOMI total-column methane measurements at a very high (50 km) spatial resolution to quantify coal and oil+gas fossil fuel emissions of methane to the atmosphere. This is critically important and valuable work, given the recent importance assigned to methane as a greenhouse gas, given that we have more power now to cut down our methane emissions than our CO₂ emissions in the short term. The fact that they report results (separately for oil+gas and coal) for roughly 100 countries/regions is nothing short of amazing, especially given some of the small areas of these countries. They also compare the oil+gas emissions to total production in order to quantify the "methane intensity" of each country, that is, the so-called "methane leakage rate". Therefore, after addressing the following comments and concerns, I suggest this work be published with all due speed.

"What are the noteworthy results?"

Their primary results are their unprecedented disaggregation of country-scale fossil fuel-based methane emissions over the time period of study. This is very noteworthy and of both high scientific and political/policy-based importance.

"Will the work be of significance to the field and related fields?"

Yes.

"How does it compare to the established literature?"

I am not a methane emissions researcher (currently), so it is difficult for me to fully answer this question. Based on the manuscript, it seems their results for various regions (Russia, China, etc) are generally consistent with previous findings.

"If the work is not original, please provide relevant references."

It certainly appears to be original.

"Does the work support the conclusions and claims, or is additional evidence needed?"

Usually, but not everywhere, as discussed in my comments below.

"Are there any flaws in the data analysis, interpretation and conclusions? Do these prohibit publication or require revision?"

Major flaw is the relatively incomplete error analysis as I discuss more fully below. This requires some level of revision.

"Is the methodology sound? Does the work meet the expected standards in your field?"

Yes.

"Is there enough detail provided in the methods for the work to be reproduced?"

It appears so, but I am not an inversion expert so I am not sure.

Response. We thank the referee for making these valuable comments. Please see our following responses.

High-Level Comments:

My central critique of this work is the relatively incomplete nature of the error analysis discussion. Given the relatively high scientific and political importance of their findings, having appropriate and valid error bars on their results is critical. First, it appears based on the “Materials and Methods” discussion that the uncertainty due to wetland emissions is not included in their posterior emissions estimate. It seems to be it must be, as TROPOMI only sees the total CH₄ emissions, even in a relatively small area. For instance, what if the entire ensemble of WetCHARTS estimates had been included, instead of just the mean? For many countries/areas this may not have been a significant contributor to the uncertainty, but certainly in some areas (especially in the tropics) it would be. The authors must discuss this potentially important source of uncertainty and why it is justified to completely ignore it.

Response. We thank the reviewer for making this good point. Each grid cell includes emissions from non-fossil fuel sources. Thus, we attribute posterior emissions to each sector based on error-weighted contributions from different sectors in the prior inventory, and the uncertainty of wetland emissions is partly taken into account.

Line 89. Here, we attribute the posterior corrections to the different sectors in the grid cell on the basis of their error-weighted contributions to the prior emissions^{23,24}, and **sectors with higher prior uncertainty are subject to larger relative corrections (Text S1). This error weighting is particularly important for grid cells containing wetland emissions, which are particularly uncertain.**

Following the reviewer’s suggestion, we have included a new set of inversion using all WetCHARTS members as the prior in the analysis.

Line 312. An additional limitation in quantifying oil-gas emissions at high latitudes and in the tropics is the uncertainty in wetland emissions. Whereas oil-gas fields at northern mid-latitudes are usually in arid regions, oil-gas fields at high latitudes and in the tropics are often co-located with wetlands and separating the two emission sectors can be difficult. Our prior estimate of wetland emissions is obtained by averaging the 9 highest-performance members of the WetCHARTs v1.3.1 inventory ensemble^{38,39}, for a global emission of 149 Tg a⁻¹ in 2019. We conducted another set of inversions using the ensemble average of all 18 members of WetCHARTs v1.3.1 with prior wetland emissions that are 10% higher globally, 20-30% higher in the tropics, and 15% lower in central Russia (Fig. S15). The resultant changes in posterior fossil fuel emissions are 0.7 Tg in Venezuela and <0.2 Tg a⁻¹ in other countries (Fig. S15).

Effects of wetland emissions on posterior methane emissions from fossil fuel exploitation

(a) WetCHARTs wetland methane emissions (high performance – all ensembles)

(b) Δ methane emissions from fossil fuel exploitation in the top 5 countries (high performance – all ensembles)

	Venezuela	Russia	India	Nigeria	Cote d'Ivoire
Posterior OG and Coal emissions using high performance WetCHART members (Tg a^{-1})	4.2	11.3	1.9	3.4	0.27
Δ emissions if using all WetCHART members (Tg a^{-1})	-0.7	0.3	-0.2	-0.15	-0.15

Fig. S15. Effects of wetland inventories on posterior fossil fuel emissions. (a) Difference in wetland methane emissions using the 9 highest-performance members and 18 all members in WetCHARTs v1.3.1^{3,5}. The five inversion domains with the highest relative difference in wetland methane emissions are shown as black rectangles. (b) The 5 top countries with the highest absolute changes of posterior methane emissions from fossil fuel exploitation if using all 18 WetCHARTs members.

Second, what is the contribution of atmospheric transport uncertainty? I realize they are doing regional inversions but still, given that they are trying to disaggregate coal, oil+gas, livestock, agricultural, and wetland emissions, I would think that atmospheric transport uncertainty could play a large role in their posterior emissions estimates. Is it feasible to perform these runs with a different transport model and compare the differences? If not, the authors MUST be clear from the outset the potential role of atmospheric transport uncertainty in potentially invalidating some of their conclusions.

Response. The reviewer makes a good point here that atmospheric transport could be an important source of uncertainty. But using a different transport model to test the difference is too expensive for us. Now we acknowledge this potential shortcoming in the discussion.

Line 326. GEOS-Chem transport error is treated as random through the observational error covariance matrix, but any systematic transport bias⁴⁰ would again propagate to inversion results.

Finally, I suggest including additional error bars in some of their figures, as discussed below.

Specific comments:

- I don't love the title. "Fuel exploitation" is not a common phrase. Suggest "fossil fuel exploitation" to be more specific.*

Response. Corrected thanks.

- Figure 2: Please consider using a log-log scale (rather than linear) to better show the agreement at the lower emission rates.*

Response. Corrected. Please check the updated Fig. 2.

Fig. 2. Methane emissions from 24 oil-gas and coal production basins across the globe. Estimates from field campaigns are compared to results from our TROPOMI inversions. For the US and Canada basins, we adjust TROPOMI’s results to campaign years using relative linear trends of observation-derived (surface measurements + GOSAT) basin-scale emissions from Lu et al.³⁷. The 1:1 line is dashed, and the correlation coefficient is shown inset. More details including references for the field campaigns can be found in Table S1-S2. **Note the log-log scale.**

• *Figure 3: Please include vertical error bars for the various top-down measurements. These are a critical part of the measurement and are typically displayed on these plots, so we can get a sense of how well the different estimates agree to within their uncertainties, and how your uncertainty (rather than just absolute value of emissions) compares. If older Top-Down uncertainty estimates are artificially low because they do not include certain contributors to uncertainty, it is important for your to add this to your discussion.*

Response. Thanks for making this good point. In Table S3, we now summarize the uncertainty of these top-down estimates. We find that only Sauniois et al. (2016, 2020) and Fraser et al. (2013) reported reasonable uncertainty ranges and other top-down uncertainty estimates are artificially low (as the referee pointed out). We feel it is difficult to display these uncertainties in Fig. 3, so we put this information in the text and refer to it in the main text.

Line 625. Top-down error estimates are either not reported or unrealistically low (<5%) (except Sauniois et al.^{3,6} and Fraser et al.⁶²), which can be found in Table S3 for more details.

Line 156. while our estimate of coal emissions ($32.7 \pm 5.2 \text{ Tg a}^{-1}$) is higher than previous top-down studies ($20\text{-}30 \text{ Tg a}^{-1}$) and in agreement with GFEI v2 (33 Tg a^{-1}) (Fig. 3c) (**details in Table S3**).

Table S3. Methane emissions from the oil-gas and coal exploitation in different top-down studies.

Top-down studies	Oil and Gas (Tg a^{-1})		Coal (Tg a^{-1})		Fossil fuel (Tg a^{-1})		Years of emissions
	Mean	Uncertainty	Mean	Uncertainty	Mean	Uncertainty	
Sauniois et al. ²⁸					101	77-126 [#]	2000-2009
					105	77-133 [#]	2003-2012
					112	90-137 [#]	2012

Saunois et al. ²⁹					101	71-151 [#]	2000-2009
					111	81-131 [#]	2008-2017
					108	91-121 [#]	2017
Turner et al. ³⁰	67	NA	30	NA	97	NA	2009-2011
Zhang et al. ³¹	59.4	2%*	21	2%*	80.4	2%*	2010-2018
Lu et al. ³²	70	NA	23	NA	93	NA	2010-2017
Maasakkers et al. ³³	67.5	<2%*	27.6	<2%*	95	<2%*	2010-2015
Qu et al. ³⁴	54	5%*	26	NA	80	5%*	2019
Fraser et al. ³⁵	NA		NA		77.8	71.8-83.8	2009
This work	62.6	50.9-73.2	32.6	29.3-37.8	95.4	81.5-108.3	2018-2020

[#]The uncertainty refers to the min-max range of reported studies.

*These studies didn't report the uncertainty specifically for the fossil fuel sector; thus, we use the uncertainty of global anthropogenic emissions here. The uncertainties in these studies are extremely low because they assume the grid-scale posterior estimates are independent with each other; thus, the uncertainty of global emissions inversely scales with the length of state vectors.

• *Figure 4: I like that you included emissions uncertainties on the TROPOMI estimates. But it is also very important to include them on the methane intensities (top plot) and emission factors (bottom plot). Please do this. (If not possible for some reason, please state why not in the text, and speculate on what your uncertainties on these values might be.)*

Response. We have added the uncertainty for the emission intensity. Please check.

Fig. 4. National methane emissions from the oil-gas and coal sectors estimated by inversion of TROPOMI observations and compared to the UNFCCC reports. The TROPOMI observations are for May 2018 – February 2020, and the UNFCCC reports are for 2019 (Annex I countries) or most recent (other countries), as compiled by the

GFEI v2 inventory of Scarpelli et al. ⁵. Iraq has not reported to the UNFCCC since 2000 and its emission is estimated in GFEI v2 using IPCC emission factors. The top 20 emitting countries are shown here; data for the 93 countries with total fuel emissions larger than 1 Gg a⁻¹ are in Table S2. Vertical bars indicate the 95% confidence levels from the inversion ensemble. The circles represent the methane intensity from oil-gas production (top panel), defined as the total oil-gas emission per unit of gas produced (assuming 90% methane content for gas) ^{4,29}, and the coal emission factor (bottom panel), defined following IPCC ³⁰ as the total coal emission per unit coal produced. **The empty rectangles denote the 95% confidence levels of oil-gas emission intensities and coal-based emission factors.** Note break in left ordinate axis of bottom panel, as Chinese coal emissions are much higher than for any other country.

• L117: Discussion of the large uncertainty on Russia's oil-gas estimate . Your data density for Russia is not much different for the U.S. it appears. It appears that the driving factor is the large disparity between GFEI v1 as compared to v2 and EDGARv6. Please comment on ways to potentially reduce this uncertainty, given the extremely important implications for these results?

Response. Here we have updated the figure about observation data frequency and results show that the US has much higher data density in oil-gas producing regions.

Figure S8. (a) TROPOMI data density (counts km⁻² a⁻¹) from May 2018 to February 2020, mapped to 0.5°×0.625° horizontal resolution. (b) Same as (a) but only for gridcells with fossil fuel methane emissions greater than 1 Gg a⁻¹.

We now discuss like this:

Line 277. It has more difficulty in Russia and the tropics (Fig. S14), where satellite observation density is relatively low and oil/gas fields are often collocated with wetlands, so that inversion results have limited information content and are sensitive to the choice of prior inventory (Table S5-S6).

Deploying facility-scale measurements and aircraft campaigns targeting the large oil-gas producing basins should be the most effective way reducing the estimate uncertainty in Russia. However, we cannot argue about this due to the high expense. But according to Jacob et al. (2022), future satellite instrument should be more effective at high latitudes. Now we say this in text.

Line 260. Future satellite instruments may be more effective at observing high latitudes and Russian point sources³¹.

• L128 area. I notice you do not comment on the ~factor of 2 discrepancy between GFEIv2 and TROPOMI for Nigeria's emissions. Please add a sentence on this, whether you think that any statements on this discrepancy are unwarranted given the relatively large uncertainty on your estimate, or if your value agrees better with GFEIv2 vs. EDGAR and possibly why.

Response. Thanks, now we say this in the main text.

Line 217. Nigeria's emissions reported to UNFCCC increased from 0.4 (GFEI v1 for 2016) to 3.3 Tg a⁻¹ (GFEIv2 for 2019) after adopting an emission factor at the upper limit of the IPCC (2006) recommendations³². Our inversion implies that the more recent report should be reduced by 40-50%.

• L131: regarding your comment on the "lower value (4.1 Tg a⁻¹) may be related to..." you should caveat this by saying " , though it is difficult to assess trends in their emissions given our the large uncertainty." Also, do the bottom-up inventories imply declining production in Venezuela during this time period, consistent with your statement? If so, say so!

Response. Yes, both EIA oil production and the bottom-up inventory show consistent declining trends in Venezuela from 2000-2019. Now we say this in text:

Line 250. Our lower value (4.0 Tg a⁻¹) may be related to **declining oil production over the 2016-2019 period as a result of intensified economic sanctions**³³ (Fig. S12).

Fig. S12. Oil production (<https://www.eia.gov/international/data/country/VEN>) and fossil fuel (oil + natural gas) methane emissions from GFEIv2 in Venezuela from 2010 to 2019.

• L133: "Is three times higher" → "Is 2.5 times higher" (3900/1500=2.6).

Response. Fixed, thanks.

• L140: "China's methane emission is" → "China's coal-based methane emissions are".

Response. Fixed, thanks.

• L141: " , lower by 10% than the UNFCCC inventory estimate" → " , consistent with the UNFCCC inventory estimate within our uncertainty." I don't think the claim that China's emissions are 10% lower is warranted given your large (17%) uncertainty.

Response. Thanks for pointing out this. Now we say this in the main text.

Line 260. China's coal-based methane emission is $18.9 \pm 3.3 \text{ Tg a}^{-1}$, **slightly but not significantly lower** than the UNFCCC inventory estimate.

Grammatical comments:

- L37: *“which can provide”* → *“can provide”*
- L63: *“We weigh”* → *“We weight”*
- L69: *“More details are in”* → *“More details are provided in”*
- L114: *China's oil-gas emissions () are two times higher than the...*
- L155: → *“...can constrain the emissions with a relative posterior uncertainty of better than 30% (2σ)...”*

Response. All fixed. Thanks!

Reviewer #4

The authors use satellite observations of methane to estimate global methane emissions from fossil fuel extraction. Overall, I think the results of this manuscript are important, novel, and worthy of publication in *Nature Communications*. I also believe the manuscript is clear and well-written. I have included several suggestions for the manuscript below.

Response. We thank the referee for making these valuable comments. Please see our following responses.

* The manuscript and SI do not include any model-data comparisons. I think that including model-data comparisons, particularly against independent CH₄ observations, is an important step for evaluating the inverse model. A handful of recent studies have found substantial differences between CH₄ emissions estimated using TROPOMI observations and GOSAT observations (see below), potentially due to errors in the TROPOMI CH₄ observations. These findings make model-data comparisons against independent observations all the more important. There are numerous independent datasets that could be used in this study for evaluation -- GOSAT observations, in situ observations from the NOAA Greenhouse Gas Reference Network, and observations from some of the campaigns cited in Fig. 2.

Response. We thank the referee for making this suggestion. Now we have included model evaluation with TROPOMI, GOSAT, and in-situ surface measurements. We have made the following changes.

Line 143. We evaluate our posterior emission estimates by implementing them in GEOS-Chem and using them to simulate column-averaged methane mixing ratios for comparison with TROPOMI and GOSAT, and surface concentrations for comparison with surface measurements from the National Oceanic and Atmospheric Administration (NOAA) network²⁶. Results show consistent improvements in the model-observation bias relative to using prior emissions (Fig. S2-S5).

Fig. S2. Bias of column-averaged methane mixing ratio in the prior and posterior GEOS-Chem simulations relative to TROPOMI from May 2018 to February 2022. To be consistent with our inversion setups, we run the GEOS-Chem model in 15 inversion domains using prior and posterior inventories at the $0.5^\circ \times 0.625^\circ$ resolution, and then aggregate the simulation results. The mean bias and root mean square errors (RMSE) are shown inset.

Fig. S3. Bias of column-averaged methane mixing ratio in the prior and posterior GEOS-Chem simulations relative to GOSAT from May 2018 to February 2022. To be consistent with our inversion setups, we run the GEOS-Chem model in 15 inversion domains using prior and posterior inventories at the $0.5^\circ \times 0.625^\circ$ resolution, and then aggregate the simulation results. The mean bias and root mean square errors (RMSE) are shown inset.

Fig. S4. Latitudinal column-averaged methane mixing ratio between GOSAT, TROPOMI, GEOS-Chem posterior runs. GEOS-Chem simulations are performed in 15 inversion domains with boundary conditions calibrated to match TROPOMI (same as our inversion setups). Thus, GEOS-Chem's latitudinal variability closely resembles TROPOMI's.

Fig. S5. Bias of surface methane concentrations in the prior and posterior GEOS-Chem simulations relative to NOAA in-situ observations inside our 15 inversion domains from May 2018 to February 2022 (ObsPack, Schuldt et al.⁴). The mean bias and root mean square errors (RMSE) are shown inset.

* A recent study by Liang et al. (<https://acp.copernicus.org/preprints/acp-2022-508/>) find differences of ~30% between inverse estimates of CH₄ for India and China that use TROPOMI observations versus GOSAT observations. Furthermore, they argue that emissions estimated using GOSAT observations are more consistent with surface observations. Liang et al. also use the updated Lorente et al. (2021) data product. By contrast, line 44 of the present manuscript argues that biases in the TROPOMI observations have largely been corrected. How do you reconcile the results of Liang et al. with the present study? Should those results imply larger uncertainty bounds on the reported CH₄ emissions estimated from TROPOMI?

Response. The reviewer makes a good point here that large differences may exist in local regions induced by the retrieval bias between TROPOMI and GOSAT. Now we acknowledge this shortcoming of our method in text.

Line 321. There are other poorly accounted sources of uncertainty in our analysis. TROPOMI observations are affected by regional bias in some parts of the world^{17,25,37} and the biases not removed by our quality flags (see Methods) would propagate to our inversion results.

According to Liang et al. (2023), the 30% difference only exists in the most eastern part of China and northern India. In other part of China where the majority of fossil fuel emissions occurs, both GOSAT and TROPOMI inversions show consistent corrections (Fig. 3 of Liang 2023). Here we present a comparison of fossil fuel emissions among different studies using GOSAT or TROPOMI or both. And we find the coal-based emissions are consistent (Table S7) in China. Now we say this in text.

Line 260. China's coal-based methane emission is $18.9 \pm 3.3 \text{ Tg a}^{-1}$, slightly but not significantly lower than the UNFCCC inventory estimate. Our estimate for China is comparable to the range of 16.2-18.0 Tg a⁻¹ from recent satellite-based estimates (Table S7)^{28,37}.

Table S7. Oil-gas and Coal emissions in China in different studies.

	Oil and Gas		Coal	
	Prior (Tg a ⁻¹)	Posterior (Tg a ⁻¹)	Prior (Tg a ⁻¹)	Posterior (Tg a ⁻¹)
This study	1.2	2.2	21.1	20.2
	1.1	2.1	18.5	18.2
	3.2	3.8	19.9	18.2
Liang et al. ³⁷	1.2	1.8	16.6	18.0
	1.2	1.4	16.6	16.2
Chen et al. ³⁸	1.1	2.7	19.5	16.6

** Materials and Methods section: How does the inverse model treat temporal variability in the emissions? The inverse model optimizes the overall magnitude and spatial patterns in the emissions, but does the inverse model optimize seasonal or year-to-year variability in the emissions? In other words, does the inverse model assume that the temporal distribution of emissions is known or fixed to the temporal distribution of the prior?*

Response. Thanks for making this good point. We assume the temporal corrections are fixed. We feel it is difficult to optimize for seasonal or monthly variability of emissions in most part of the world. We now make it clear in text.

Line 323. High latitudes and tropics have large seasonal variations in observation density that would affect inversion results if fossil fuel emissions were seasonally variable (**we assume that they are not**). Independent inversions for different seasons show near-zero posterior corrections in the wintertime at high latitudes because of the low observation density (Fig. S8-S9, S16, Text S2).

Text S2. Seasonal posterior correction factors and effects of data density at high latitudes

We also calculated posterior emissions from fossil fuel exploitation using TROPOMI observation in different seasons at northern high latitudes, where the observation density varies considerably across one year (Fig. S8-S9). In those high-latitude countries, posterior corrections are nearly zero in winter because of scarce observations and low averaging-kernel sensitivities (Fig.S16). **This result also demonstrates that seasonal corrections are partly influenced by satellite data availability, which does not necessarily reflect the temporal variability in emissions in regions with uneven observation density.** In this study, we do not try to optimize for higher temporal variability of emissions because uneven and inadequate seasonal sampling frequencies are present in most regions of the world (Fig. S8-S9).

Figure S8. (a) TROPOMI data density (counts $\text{km}^{-2} \text{a}^{-1}$) from May 2018 to February 2020, mapped to $0.5^\circ \times 0.625^\circ$ horizontal resolution. (b) Same as (a) but only for gridcells with fossil fuel methane emissions greater than 1 Gg a^{-1} .

Figure S9. Fraction of TROPOMI observation in different seasons (DJF, MAM, JJA and SON), mapped to $0.5^\circ \times 0.625^\circ$ horizontal resolution. Gridcells with zero observation are shown as white.

Fig. S16. Posterior correction factors in different seasons at high latitudes north of 30°N. This test uses GFEIv2 as the prior inventory. Please note that the seasonal corrections here are largely determined by the sampling frequency in different seasons (Fig. S8-S9), which do not necessarily reflect the temporal variability in emissions. See Text S2 for discussion.

** Lines 18, 49, and 161: I find the wording in these lines somewhat misleading. The inverse model, as described in the Materials and Methods section, is used to estimate emissions at 50km resolution in selected grid boxes of the model domain. By contrast, the authors aggregate model grid boxes into coarser regions for other areas of the model domain. Hence, I think it's fair to say that the inverse model is used to estimate emissions at spatial resolutions up to 50km resolution, but the emissions are not optimized at that resolution everywhere.*

Response. Right. Now we make it clear that we only solve the emissions at 50km resolution for the fossil fuel emission sector.

Line 18. Here we use 22 months (May 2018-Feb 2020) of satellite observations from the TROPOMI instrument to better quantify national emissions worldwide by inverse analysis at **up to 50 km resolution**.

Line 56. Here we conduct a global ensemble of regional inversions of TROPOMI data to quantify emissions from fossil fuel exploitation worldwide at **up to 50-km resolution**,

Line 83. We conduct inversions of the TROPOMI observations in each domain from May 2018 to February 2020 to quantify emissions with a resolution of **up to 0.5°×0.625° (~50km)** for grid cells with significant fuel emissions (>1 Gg a⁻¹).

Line 337. In summary, we have used 22 months of TROPOMI satellite observations (May 2018 – February 2020) in an inverse analysis to quantify methane emissions from the fossil fuel industry (oil, gas, and coal) globally at **up to 50 km resolution**.

Lines 36-38: This sentence is grammatically complex, and I think simplifying or breaking up the sentence would improve readability.

Response. Thanks, now we say this in text.

Line 46. Top-down approaches apply inverse methods to infer emissions from measurements of atmospheric methane. They use prior information from the bottom-up inventories and provide an independent way of improving these inventories.

** Lines 46-47: I also find these lines to be somewhat misleading, and I recommend deleting this sentence from the manuscript. It is true that existing, global inverse modeling studies often estimate emissions at coarse spatial resolutions. With that said, the present study also makes compromises on resolution to improve the computational feasibility of the inverse model. For example, the emissions in the present study are estimated at coarser, aggregate resolutions in grid boxes where fossil fuel emissions are less than 1 Gg a-1. In addition, my impression is that the inverse model in the present study does not correct temporal variability in the emissions. Overall, the state vector in this study has 5651 elements, fewer elements than many global GHG inverse modeling studies that use satellite data.*

Response. Yes, we agree with the referee here. We now have deleted that sentence. Thanks!

Line 76: *Independent estimates of what?*

Response. Now we deleted 'independent' because this word may cause some confusion.

REVIEWERS' COMMENTS

Reviewer #1 (Remarks to the Author):

The author has made a commendable effort to adequately address my concerns and suggestions. I have no further comments and suggest the work for the final publication.

Reviewer #4 (Remarks to the Author):

I think the authors have done an excellent and thorough job of revising the manuscript. I have no further comments and recommend publication.

Response to referee comments on “Worldwide quantification of national methane emissions from fuel exploitation using high-resolution inversions of satellite data”

Reviewer #1 (Remarks to the Author):

The author has made a commendable effort to adequately address my concerns and suggestions. I have no further comments and suggest the work for the final publication.

Reviewer #4 (Remarks to the Author):

I think the authors have done an excellent and thorough job of revising the manuscript. I have no further comments and recommend publication.

Response. We thank the two referees for their careful reading of the revised manuscript. Because both referees do not have further comments, we submit the last version of manuscript after incorporating the editor’s suggestions.